# COVID-19 testing, timeliness and positivity from ICMR's laboratory surveillance network in India: Profile of 176 million individuals tested and 188 million tests, March 2020 to January 2021

Manickam Ponnaiah[1☯], Rizwan Suliankatchi Abdulkader[1☯], Tarun Bhatnagar[1], Jeromie Wesley Vivian Thangaraj[1], Muthusamy Santhosh Kumar[1], Ramasamy Sabarinathan[1], Saravanakumar Velusamy[1], Yogesh Sabde[2], Harpreet Singh[3], Manoj Vasanth Murhekar[1]*

1 ICMR National Institute of Epidemiology, Chennai, Tamil Nadu, India, 2 ICMR National Institute for Research in Environmental Health, Bhopal, Madhya Pradesh, India, 3 Division of Biomedical Informatics, Indian Council of Medical Research (ICMR), New Delhi, India

☯ These authors contributed equally to this work.
* mmurhekar@gmail.com

## Abstract

### Background

The Indian Council of Medical Research set up a pan-national laboratory network to diagnose and monitor Coronavirus disease 2019 (COVID-19). Based on these data, we describe the epidemiology of the pandemic at national and sub-national levels and the performance of the laboratory network.

### Methods

We included surveillance data for individuals tested and the number of tests from March 2020 to January 2021. We calculated the incidence of COVID-19 by age, gender and state and tests per 100,000 population, the proportion of symptomatic individuals among those tested, the proportion of repeat tests and test positivity. We computed median (Interquartile range—IQR) days needed for selected surveillance activities to describe timeliness.

### Results

The analysis included 176 million individuals and 188 million tests. The overall incidence of COVID-19 was 0.8%, and 12,584 persons per 100,000 population were tested. 6.1% of individuals tested returned a positive result. Ten of the 37 Indian States and Union Territories accounted for about 75.6% of the total cases. Daily testing scaled up from 40,000 initially to nearly one million in March 2021. The median duration between symptom onset and sample collection was two (IQR = 0,3) days, median duration between both sample collection and

**Data Availability Statement:** The data underlying the results presented in the study are available from the Indian Council of Medical Research (ICMR), New Delhi, https://main.icmr.nic.in/content/contact-us. Data cannot be shared publicly because it is restricted by the ethics committee that approved the study on the grounds that the raw data contains identifying information such as names, phone numbers and addresses. Data are available from the Secretary to the Government of India & Director General of ICMR (email address: secy-dg@icmr.gov.in) for researchers who meet the criteria for access to confidential data.

**Funding:** The author(s) received no specific funding for this work.

**Competing interests:** The authors have declared that no competing interests exist.

testing and between testing and data entry were less than or equal to one day. Missing or invalid entries ranged from 0.01% for age to 0.7% for test outcome.

## Conclusion

The laboratory network set-up by ICMR was scaled up massively over a short period, which enabled testing a large section of the population. Although all states and territories were affected, most cases were concentrated in a few large states. Timeliness between the various surveillance activities was acceptable, indicating good responsiveness of the surveillance system.

## Introduction

The coronavirus disease 2019 (COVID-19) pandemic caused by the novel severe acute respiratory syndrome-coronavirus-2 (SARS-CoV-2) infection has presented a monumental challenge to India's health and socio-economic situation and has caused a level of disease burden next only to the United States [1]. As the second most populous country with 37 states and Union Territories (UTs), the pandemic has severely impacted India. The key public health strategy towards reducing the transmission of COVID-19 is testing suspected and high-risk individuals followed by tracking and contact tracing. This strategy relied mainly on laboratory confirmation of COVID-19 by real-time reverse transcription-polymerase chain reaction (RT-PCR) among suspected and at-risk groups.

In bringing the COVID-19 testing strategy to operation, the Indian Ministry of Health and Family Welfare established a network of COVID-19 testing laboratories under the national medical research body, the Indian Council of Medical Research (ICMR). The research body was entrusted with framing the testing guidelines, setting up a network of COVID-19 testing labs, accrediting laboratories for testing, issuing guidelines, and approving testing kits. ICMR's COVID-19 laboratory surveillance network is one of India's most extensive networks for communicable diseases. Initially started in February 2020 with a small number of national and regional labs, the network has grown to include more than 2300 government and private laboratories as of March 2021. This network maintains data on individuals who have been tested for COVID-19 in ICMR-approved laboratories in the country.

India's COVID-19 testing strategy evolved with the changing needs of the pandemic. In March 2020, when the testing resources were limited, only high-risk symptomatic individuals, international inbound travellers, high-risk contacts and patients with Severe Acute Respiratory Illness (SARI) were eligible for testing. During the later phases of the pandemic, the eligibility criteria were expanded to include asymptomatic contacts, surveillance of symptomatic persons in containments zones, patients with Influenza-Like Illness (ILI) and, migrant workers. Since September 2020, patient-initiated testing was also added to the criteria, along with situations such as international travel, screening before undergoing medical/surgical procedures were also eligible for testing. The government took several public-centric initiatives to provide free and comprehensive access to testing. Persons who wanted testing were asked to contact the nearest public health care facility where testing was provided free of cost. Testing was carried out at homes of symptomatic patients/contacts of laboratory-confirmed cases by healthcare workers during routine house-to-house fever surveillance. Mobile vans located at strategic places in major towns and cities also provided testing facilities to people who wanted to be

tested. Apart from these government initiatives, many private labs and hospitals also provided fee-based testing [2, 3].

Earlier attempts at describing the epidemiology of COVID-19 patients in India has been limited to single centre experiences from teaching hospitals [4–6]. Our group had previously published descriptive epidemiological data on ten million tested individuals [7]. With more than 220 million samples tested as of March 2021 and the evolution of the pandemic, analysis of data from this network can provide the epidemiological profile for affected individuals and the characteristics of the pandemic [8] and guide control measures. Though India is one country, every state within it and every district within the states function as independent entities when it comes to the health of its population [9].

In this context, based on the ICMR COVID-19 laboratory surveillance network, we described the time and place distribution of persons tested for COVID-19 and documented the quality, timeliness, and data quality of the surveillance system.

## Methods

### Data source

The data reported here is of the individuals tested at the laboratories and testing centres accredited to the ICMR COVID-19 diagnosis and surveillance network. This data was collected through a mobile application developed for this purpose by ICMR. The form used to collect data was called the "ICMR Specimen Referral Form for COVID-19 (SARS-CoV2)" [10]. The laboratory personnel interviewed the individual at sample collection using a print version of the form. Information was collected on name, age, sex, address, contact number, whether follow up sample, type of specimen (nasal, throat, nasopharyngeal or others), type of patient category (whether it was for routine surveillance in containment/non-containment zones, Severe Acute Respiratory Illness/Influence Like Illness patients in hospitals, high-risk asymptomatic/symptomatic contacts, pregnant women, symptomatic neonates or patient-initiated situations like travel), clinical signs and symptoms (like cough, breathlessness, sore throat, diarrhoea, nausea, chest pain, vomiting, haemoptysis, nasal discharge, fever, body ache, sputum, abdominal pain), co-morbidities (like diabetes, hypertension, malignancy, chronic lung, kidney or liver diseases or immunocompromised conditions), and hospitalisation details (if applicable). Each reporting centre/laboratory manually entered the above information in the mobile/web application on the day of sample collection. All laboratories in the network used one of the ICMR approved Reverse Transcriptase–Polymerase Chain Reaction (RT-PCR) kits or Rapid Antigen Test (RAT) kits for detection of SARS-CoV-2 infection. When RT-PCR kits were used for testing, the laboratories were required to enter the individual genes' results and an overall positive or negative result. The algorithms for declaring an RT-PCR test as positive were dependent on the type of kit used. Early in the pandemic, when only one approved kit was available, the testing happened in two stages–if the 'E' gene was not detected, the test was declared negative. If the 'E' gene was detected, 'ORF' and 'RdRp' genes were tested, and the result was given as positive if one of them was detected and as negative otherwise. At present, all kits are multiplex assays where are three genes are tested, and a positive result is declared if any two genes are detected with a cycle threshold value of <35 and as negative otherwise. When RAT kits were used, only an overall positive or negative result was entered. Each record was updated with the test outcome whenever the results became available.

### Data reference period

We analysed data from March 1, 2020 to January 31, 2021.

## Description of the ICMR COVID-19 laboratory surveillance network database

To monitor the evolution and spread of the pandemic in real-time, 2294 laboratories (as of January 2021) and testing centres were required to enter their records via a web application designed specifically for this purpose. The centralised database was managed at the ICMR headquarters in New Delhi. The database had an individual-level table containing personal characteristics and a test-level table containing the test characteristics in MySQL. All reporting units in the network entered data daily from every district.

## Data management

We extracted data from the database within the reference period. We linked the individuals tested to their results employing a unique ID with separate unique IDs used for multiple testing of an individual. After extraction, we cleaned the data using range checks for dates and continuous variables and legal options for categorical variables. The detailed workflow of data cleaning is shown in S1 Fig.

## Statistical analysis

We analysed the data at the individual and test levels by person, time, and place. At the individual level, we calculated the number of positive cases by age, gender, State/UT and ISO (International Organization for Standardization) week. Similarly, we calculated persons tested and incidence (log-transformed) per 100,000 population and percentage of individuals who tested positive. We projected the 2020 population denominators for states and districts using the decadal growth rates from the two past censuses.

We generated epi-curves for India and a few major states and weekly per cent change over ISO weeks. We constructed epidemic curves based on the dates of laboratory confirmation (as entered in the database), symptom onset, and sample collection. Epi-curves were generated for selected states across three burden categories based on total positive cases (<100,000, 100,000 to 400,000 and >400,000) and compared with that of epi-curve at the national level.

We created area maps at the state level and cumulative incidence by ISO week at the district level to depict persons tested and positive cases per 100,000 population and percentage positive among tested.

At the level of the tests, we calculated tests per 100,000 population, % symptomatic among tested, % repeat tests, and test positivity by ISO week. We captured the increase in the testing capacity of the surveillance network across three scale-up phases (Phase I–March to May 2020, Phase II–June to August 2020, and Phase III–beyond August 2020). In order to describe the timeliness of the system, we calculated time-interval for three categories: symptom onset to sample collection, collection to testing and testing to data entry. We calculated the median [Inter Quartile Range (IQR)] of these time intervals by scale-up phase, test outcome and location. We conducted all analyses using R server edition version 4.0.3 and QGIS version 3.16.

## Research ethics

We performed the analysis on de-identified data. Individuals with access to the database did not retrieve any individual addresses, phone numbers, or names. The ICMR-COVID-19 National Ethics Committee (CoNEC) approved the study.

**Table 1. COVID-19 testing, incidence and positivity by gender and age among tested individuals, India (March 2020 to January 2021).**

| Indicators | Overall | Age group (in years) | | | | Gender | |
|---|---|---|---|---|---|---|---|
| | | 0–17 | 18–40 | 41–60 | >60 | Male | Female |
| Total population (in millions) | 1402.3 | 517.2 | 538.3 | 246.8 | 100.0 | 721.8 | 680.5 |
| Persons tested per 100,000 population | 12583.9 | 4549.0 | 18152.4 | 16937.9 | 13799.2 | 14642.1 | 10392.5 |
| Incidence per 100 population | 0.8 | 0.2 | 1.0 | 1.3 | 1.5 | 0.9 | 0.6 |
| Percentage of persons positive among tested | 6.1 | 3.6 | 5.2 | 8.4 | 10.9 | 6.4 | 5.7 |

## Results

### Testing and incidence by age, gender and location

During the reference period, 176 million individuals and 188 million tests were recorded in the surveillance system. Per 100,000 people, the number tested was 12,583 overall, 14,642 among males, and 10,392 among females. The overall incidence of COVID-19 per 100 individuals was 0.8 (0.9 among males and 0.6 among females). The highest incidence (1.5 per 100 population) was in those >60 years compared to 0.2 in those 0–17 years, 1.0 in those 18–40 years and 1.3 in those 41–60 years. 6.1% of the persons tested were positive for SARS-CoV-2, with the highest positivity (10.9%) among those above 60 years (Table 1). The southernmost and northernmost states had more persons tested per 100,000 population than the central, western and north-eastern regions. Delhi ranked first with 47,610 persons tested per 100,000 population, whereas Uttar Pradesh, Madhya Pradesh, Rajasthan and West Bengal had the lowest numbers among the larger Indian States (Fig 1A, S1 Table). The spatial distribution of incidence was similar to that of testing, with the striking exception of Maharashtra, which showed high incidence and positivity (where testing was moderate) and Telangana, which showed low incidence and positivity (where testing was higher). The highest incidence of COVID-19 in the population was seen in Goa (3.7%) and Delhi (3.3%), and the highest test positivity was seen in Maharashtra (15.5%) and Goa (13%) (Fig 1B and 1C, S1 and S2 Tables). Detailed district-level results can be found at https://sarizwan.shinyapps.io/ICMRCOVID_RIZWAN/.

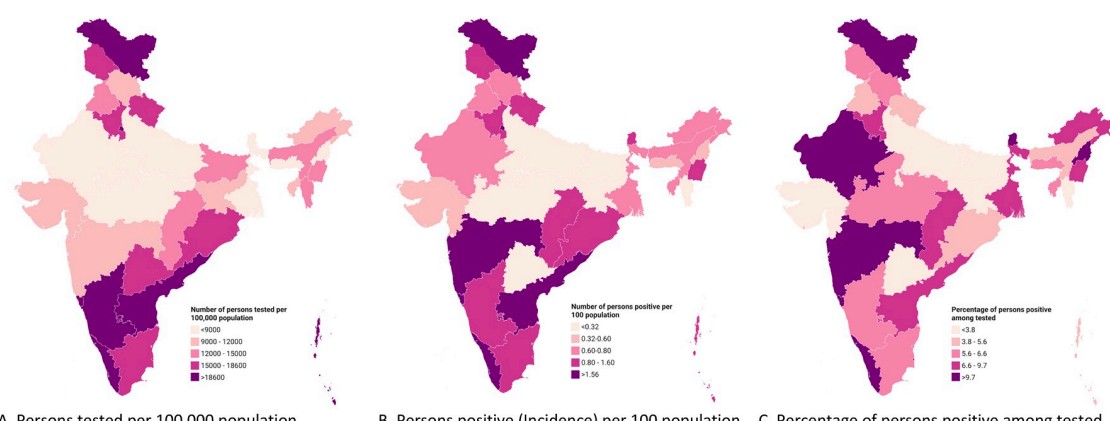

A. Persons tested per 100,000 population B. Persons positive (Incidence) per 100 population C. Percentage of persons positive among tested

**Fig 1. State-wise distribution of COVID-19 testing, incidence and positivity among tested individuals, India (March 2020 to January 2021).**

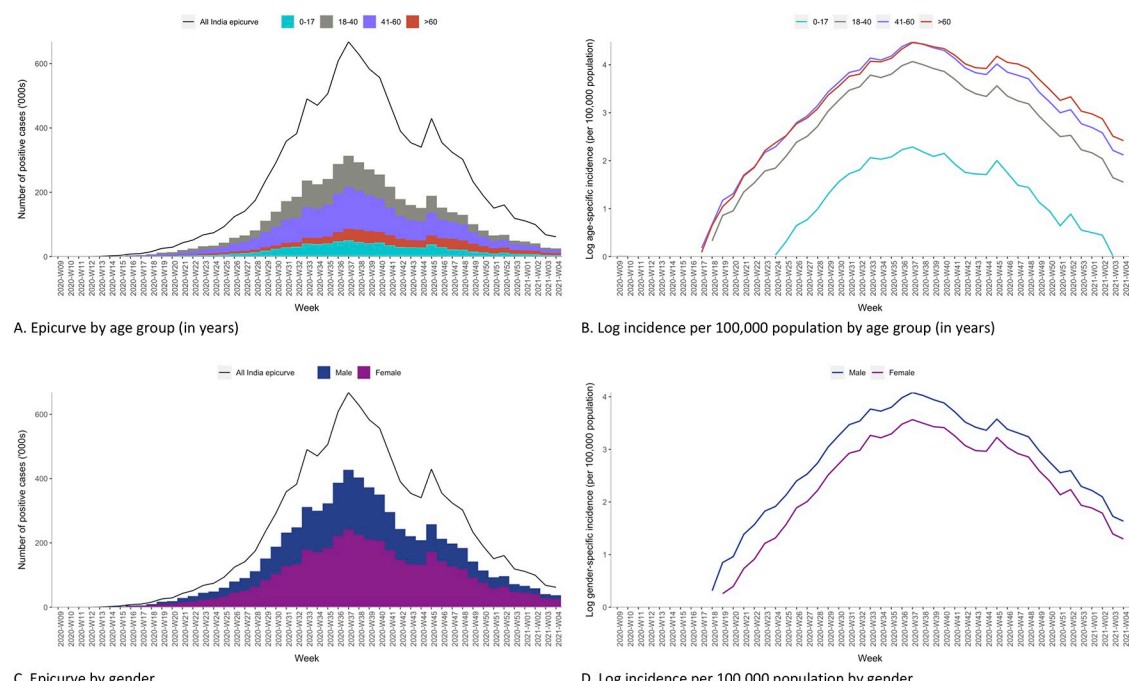

**Fig 2. Distribution of laboratory-confirmed COVID-19 cases by week and demographic characteristics, India (March 2020 to January 2021).**

## Time trends in incidence and epi-curves by age, gender and location

Over time, cases increased steadily, reaching a peak during week 37 in 2020 and then briefly declined, then again increased reaching a second (lower) peak around week 45, before continuing a prolonged steady decline (Fig 2A). Incidence was consistently higher among those aged >60 and males throughout the reference period, whereas the absolute caseload was primarily attributed to those aged 18–40 years and males (Fig 2A–2D). Epi-curves by states indicated that the early phase of the pandemic was driven predominantly by Maharashtra, Andhra Pradesh, Karnataka, Tamil Nadu, Delhi and Uttar Pradesh. In contrast, the latter part was driven by the states of Kerala, Delhi, Rajasthan, West Bengal, Odisha, Haryana, Uttarakhand, Goa, Himachal Pradesh and Manipur (Fig 3A and S2 Fig). Kerala reported the highest weekly increase (by 271 times) in terms of absolute case count between ISO weeks 51 and 52 (Fig 3B). In contrast, states like Maharashtra, Haryana, Madhya Pradesh, Uttarakhand, Goa and Himachal Pradesh showed a bi-modal distribution with a tri-modal distribution observed in Delhi and Gujarat (S2 Fig).

Examination of the incidence across time and space at the district level revealed that southern and coastal states like Kerala, Tamil Nadu, Maharashtra, Karnataka, Orissa, and Ladakh and a few districts in the North-East were the first to be affected. All districts reported at least one case for the first time during the week ending on 16 August 2020 (ISO week 33) (Fig 4 and S1 and S2 Videos). Telangana was an outlier compared to other southern states as it reported fewer cases with time, with half of its districts reporting no cases towards the end of the observation period (Fig 4II.A, 4II.B and 4III.A). At the end of the reference period (Fig 4III.A), when most parts of India were still reporting fewer than 3 cases per week per 100,000 individuals, the whole of Kerala and a few pockets in western and northeastern Maharashtra were reporting >20 cases per 100,000 individuals.

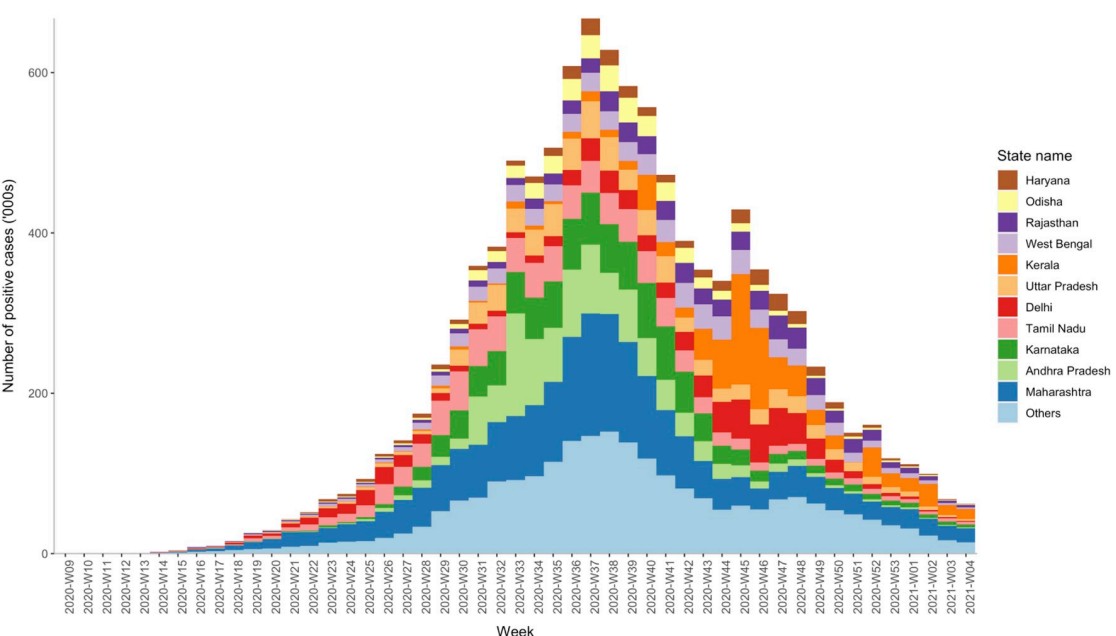

A. Number of positive cases

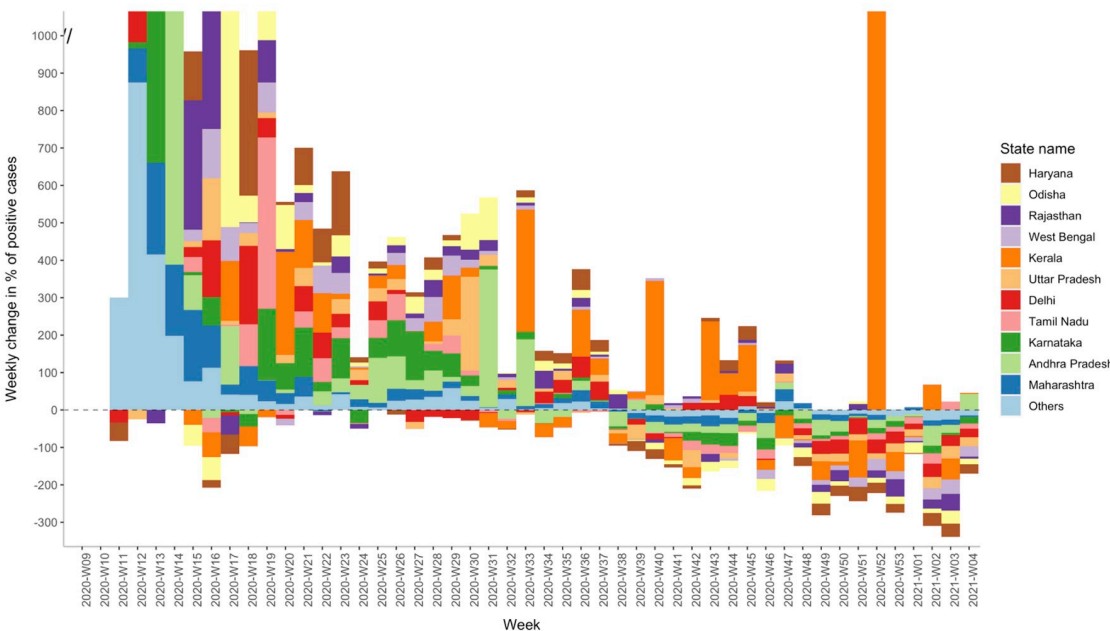

B. Percentage change in weekly positive cases

**Fig 3. Distribution of laboratory-confirmed COVID-19 cases and percentage change by week and the state of residence, India (March 2020 to January 2021).**

The epi-curve based on confirmation date was shifted more to the right than that based on sample collection date, even though the peak weeks did not differ between these two curves (Fig 5A). However, we noted an unusually high number of cases confirmed on week 45, whereas this was not the case with the sample collection curve. Examination of the state epi-

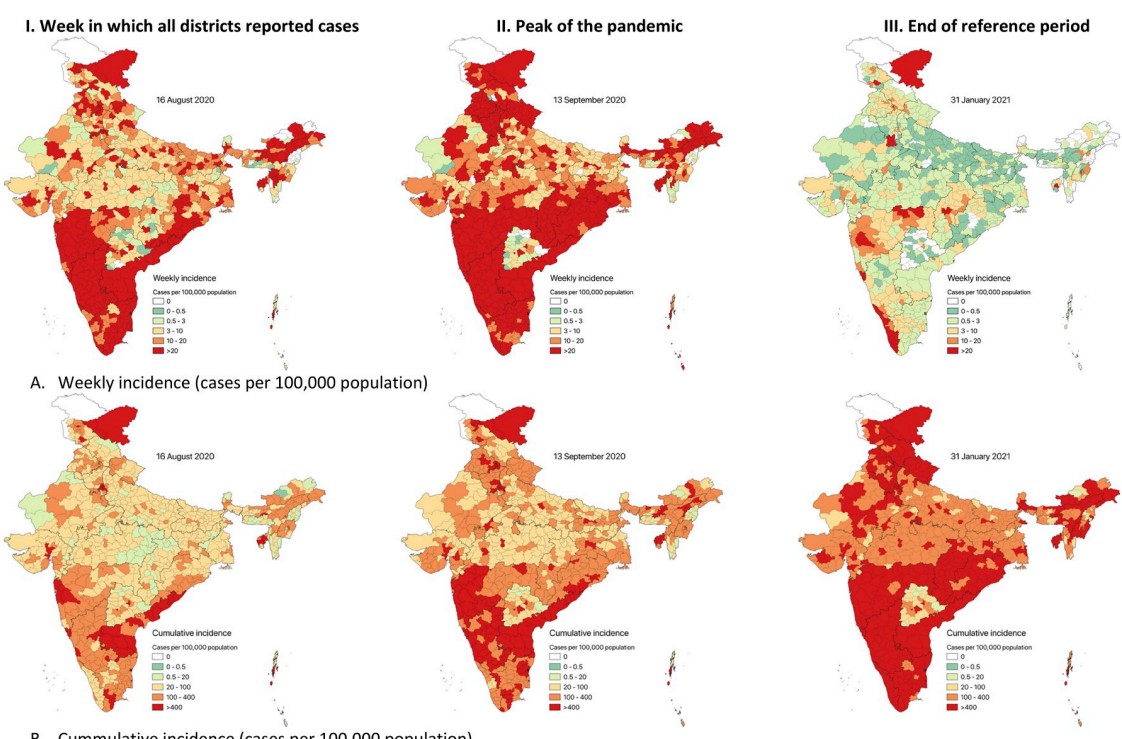

**I. Week in which all districts reported cases** **II. Peak of the pandemic** **III. End of reference period**

A. Weekly incidence (cases per 100,000 population)

B. Cummulative incidence (cases per 100,000 population)

**Fig 4. Weekly and cumulative incidence of laboratory-confirmed COVID-19 cases by the district of residence, India (March 2020 to January 2021).**

curves showed that Kerala declared many cases on week 45, coinciding with a smaller second peak in the national curve, which may be due to cumulative reporting of a substantial number of cases in week 45 (S2 Fig). When drawn with dates of symptom onset, sample collection, and confirmation, the epi-curves for symptomatic cases show that the curve based on symptom onset date predated the other two curves by a week (Fig 5B).

The number of tests per 100,000 population reached an aggregate of 13,431 at the national level during the reference period. Of the tests conducted, 7% were among symptomatic individuals, 5.1% were repeat tests, 6% were positive, and 0.7% had no outcomes reported (Table 2). Scale-up of the testing facilities from just a single laboratory in January 2020 to that of more than 2300 labs in January 2021 was reflected in the steady increase of daily tests from 40,278 in phase I and 405,966 in phase II to that of 962,716 tests in phase III (Table 2). However, the frequency of daily tests declined slightly towards the end of the reference period in January 2021. Excluding the initial high proportions, the trend of test positivity increased up to week 31, after which it declined steadily. Similarly, the percentage of tests carried out among symptomatic individuals was very high up to week 29, after which it declined. The percentage of repeat testing remained relatively constant throughout the reference period but declined towards the end, coinciding with the decline in overall testing (Fig 6).

Delhi had the highest testing proportion, with 51,414 tests per 100,000 individuals. Telangana (24.1%) and Gujarat (18%) performed the most testing among symptomatic persons. The state with the highest proportion of repeat testing was Andhra Pradesh (11.8%), followed closely by Nagaland (9.1%) and Uttar Pradesh (8.6%). Except for Arunachal Pradesh and Lakshadweep, all other states and UTs tested more in phase III than earlier (S3 Table).

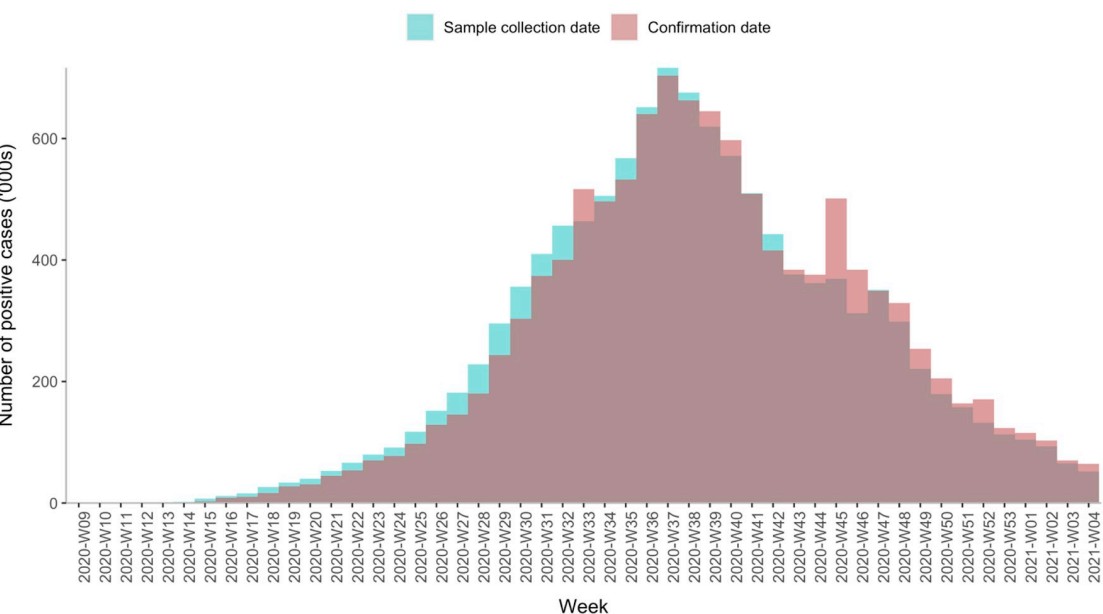

A. Epicurve by sample collection and confirmation dates (among all positive cases)

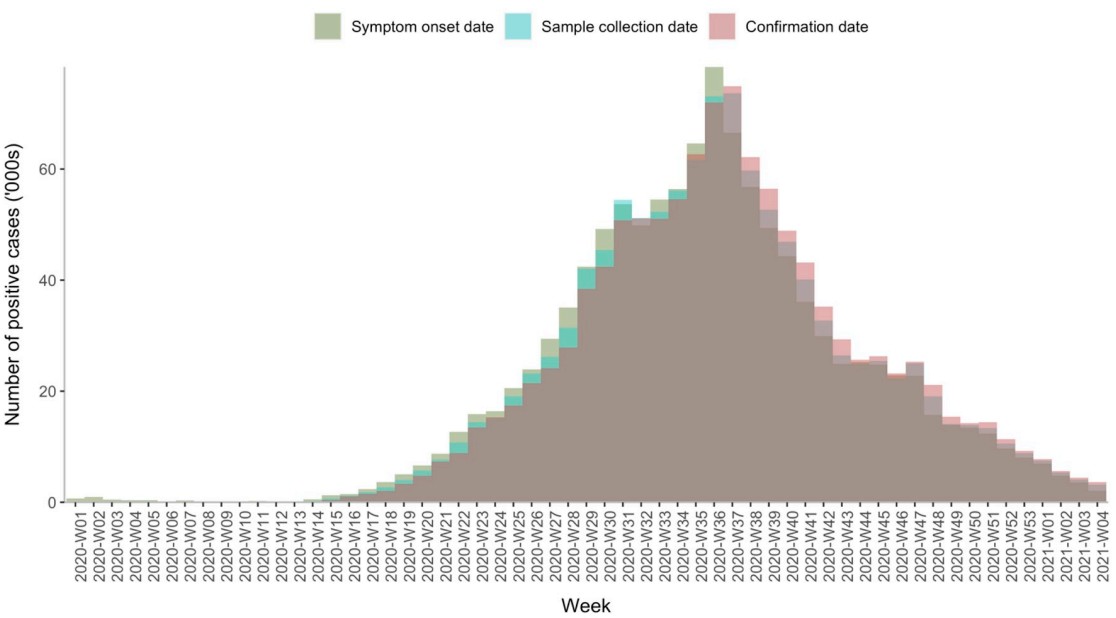

B. Epicurve by symptom onset, sample collection and confirmation dates (among symptomatic positive cases)

**Fig 5. Distribution of laboratory-confirmed COVID-19 cases by dates of symptom onset, sample collection and result confirmation, India (March 2020 to January 2021).**

## Timeliness of the surveillance network

The Median (IQR) duration between symptom onset and sample collection was 2 (IQR: 0 to 3) days, and that between sample collection and testing and between testing and reporting were 0 (IQR: 0 to 1) days. There was no difference in the duration between a negative and positive

**Table 2. Characteristics of the COVID-19 tests carried out in the laboratory surveillance network, India (March 2020 to January 2021).**

| Indicators | Value |
|---|---:|
| Total number of tests (in million) | 188.3 |
| Tests per 100,000 population | 13,431.2 |
| % of individuals who were symptomatic when tested | 7.0 |
| % of tests that were repeat tests | 5.1 |
| Outcome of tests | |
| % Positive | 6.0 |
| % Negative | 93.3 |
| % Without result | 0.7 |
| Phase-wise tests | |
| No. of tests per day (%) in Phase I | 40,278 (2.0) |
| No. of tests per day (%) in Phase II | 405,966 (19.8) |
| No. of tests per day (%) in Phase III | 962,716 (78.2) |

Note: Phase I– 01-03-2020 to 31-05-2020, Phase II– 01-06-2020 to 31-08-2020, Phase III–>01-09-2020.

test, except for a longer IQR (1 to 4 days) for the symptom to collection time for positive tests compared to (IQR: 1,3) for negative tests. Across the scale-up phases, the symptom to collection duration decreased from 3 days to 2 days, and the other intervals decreased from 1 day to zero-day with narrower inter-quartile ranges (Table 3, Fig 7).

Among the states, symptom to collection duration varied from zero to 3.5 days and collection to testing duration was typically less than one day. In contrast, the testing to entry time was longer for Andaman and Nicobar Islands (13 days, IQR: 2 to 26), Kerala (10 days, IQR: 2 to 22) and Nagaland (4 days, IQR: 1 to 12). In general, union territories and north-eastern and

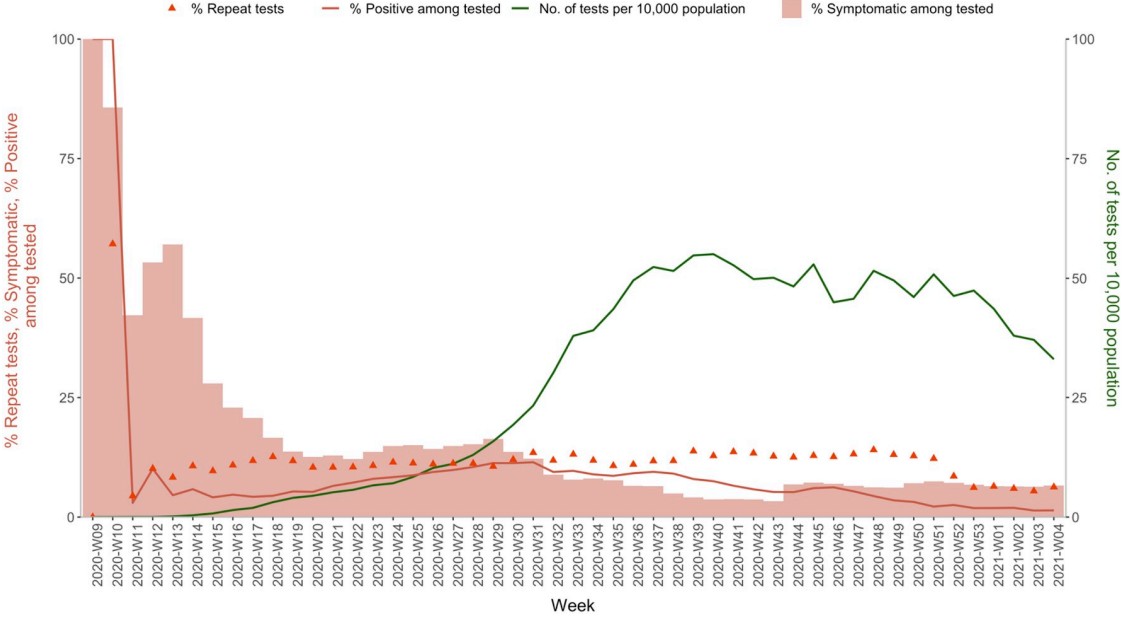

**Fig 6. Trends of the characteristics of the COVID-19 tests, India (March 2020 to January 2021).**

**Table 3. Timeliness of the COVID-19 tests carried out in the laboratory surveillance network by scale-up phases and test results, India (March 2020 to January 2021).**

| Days taken [Median (IQR)] | Overall | Test result | | Scale-up phase | | |
|---|---|---|---|---|---|---|
| | | Negative | Positive | Phase I | Phase II | Phase III |
| Symptom onset to sample collection | 2 (0, 3) | 2 (0, 3) | 2 (1, 4) | 3 (1, 5) | 2 (1, 4) | 2 (0, 3) |
| Sample collection to testing | 0 (0, 1) | 0 (0, 1) | 0 (0, 1) | 1 (0, 2) | 1 (0, 2) | 0 (0, 1) |
| Sample testing to data entry | 0 (0, 1) | 0 (0, 1) | 0 (0, 1) | 1 (0, 3) | 1 (0, 2) | 0 (0, 1) |

Note: Phase I– 01-03-2020 to 31-05-2020, Phase II– 01-06-2020 to 31-08-2020, Phase III–>01-09-2020.

island states experienced longer durations than large states for positive test results and across all scale-up phases. Positive tests were generally entered more or as quickly as negative tests, except for Nagaland and Telangana. Nagaland took the longest time (7 days, IQR: 3 to 10) to enter positive tests, followed by Kerala (5 days, IQR: 1 to 15) and Andaman and Nicobar Islands (4 days, IQR: 1 to 21 days) (S4–S6 Tables).

## Missingness of data

A database of this size is expected to have missing data. We found that missingness ranged from 0.01% for the age variable to 0.7% for the test result outcome. The date of testing was missing in 0.08% of the entries.

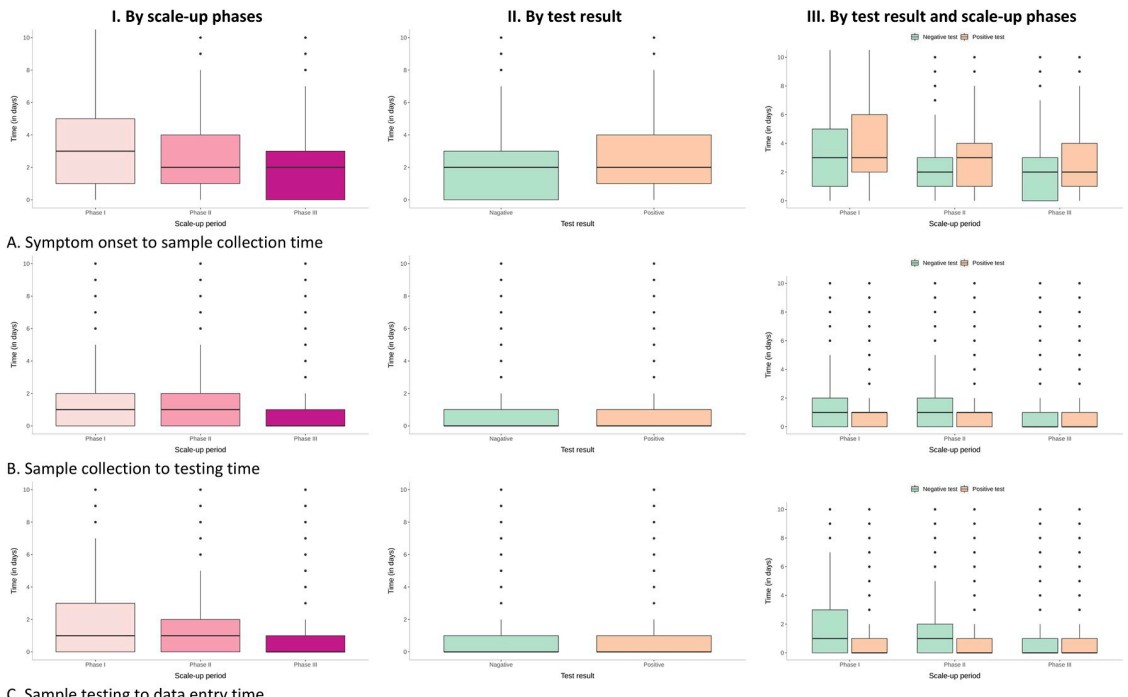

**Fig 7. Timeliness of the COVID-19 tests carried out in the laboratory surveillance network by scale-up phases and test results, India (March 2020 to January 2021).**

## Discussion

In the context of India's COVID-19 pandemic, we analysed the data on 176 million individuals tested in 2294 laboratories of the countrywide ICMR COVID-19 surveillance network during the period March 2020 to Jan 2021. Through this analysis, we profiled the descriptive epidemiology of COVID-19 in India and documented performance-related indicators of an extensive network of laboratory surveillance.

The cumulative incidence during the reference period was 0.8 per 100 individuals, which was 105th in the world, similar to countries like Malaysia, Saint Vincent and the Grenadines, Finland, El Salvador, Guatemala, Botswana and Nepal [11]. Testing per 100,000 individuals in India was 13,431, ranking 156th globally and comparable to countries like Morocco, South Africa, Iraq, and Malaysia [12]. Test positivity percentage was 6% which was about 50th globally and comparable to countries like Morocco, Kuwait, Togo, Cote d'Ivoire, Qatar, Ireland and France [13]. The overall percentage of symptomatic individuals tested was lower in our study due to two main reasons. Firstly, a large proportion of testing was carried out among asymptomatic individuals who were contacts of cases or who were picked up for random testing in containment areas [14, 15] and secondly, data on symptoms may have been inaccurately reported, collected, or entered in the database.

Many known epidemiological facts about the COVID-19 pandemic, such as higher incidence among males and the elderly was observed in the Indian context [16, 17]. The higher incidence among the elderly was probably due to their ageing immunological and physiological status and the presence of co-morbidities such as diabetes and hypertension [18, 19]. It could also be explained by the health-seeking behaviour of older adults compared to young people. Young people tend to have milder symptoms and to have less interest in getting tested. This can be seen in serologic studies where the seropositivity is comparable between older and younger adults [20]. Similarly, the higher incidence in males could be attributable to biological (such as higher expression of angiotensin-converting enzyme-2 receptors), sociological (such as poorer lifestyle choices and riskier behaviours) factors and cultural factors (such as higher access to testing facilities and importance given to men when it comes to health-seeking) in the Indian context [21]. Further, it was seen that the testing rate was higher in men than in women across all states, in contrast to high-income countries in the west [22]. Among those tested, the proportion of women was highest in Andhra Pradesh (46.4%) and lowest in Dadra and Nagar Haveli (23.4%). It has been previously shown that young women in certain states of India were less likely than men to practice key COVID-19 preventive behaviours [23]. Thus, such a low frequency of testing behaviour among women was also reflected in low prevalence among women.

The spatial distribution of COVID-19 incidence in India, where certain states like Maharashtra, Delhi, Tamil Nadu, Andhra Pradesh, Kerala and Karnataka, were severely affected in the early stages of the pandemic, could be explained by the high level of urbanisation and population density as compared to the other states [24, 25]. The distribution in Ladakh was unique since it is a tourist destination, and imported cases could have influenced the peak in the initial phase. The relationship between the level of testing and positivity and incidence has been established in previous studies [12, 26]. When the testing levels are relatively high, the incidence and test positivity are low and vice versa, assuming a constant transmission. In our study, the exact relationship was found except in Telangana and Maharashtra. However, the possibility of high levels of under-ascertainment of cases within the states cannot be overstated.

Periodic changes in the testing strategy at different pandemic phases were reflected in the upward trends in the frequency of asymptomatic individuals and the declining trends in repeat

testing [27]. Initially, when the testing strategy was exclusively done among the high-risk category, the percentage of symptomatic individuals tested was high. However, later on, when the criteria were made more inclusive, the same indicator tapered off to around 7–10% Changes in testing criteria were also reflected in the percentage of repeat testing of the individuals. During the early phases, tests were repeated every week for hospitalized patients, and a negative test was mandatory before discharge. When this criterion was relaxed, the repeat testing proportions tapered to around 5%. The time trend in test positivity reflected the increased levels of transmission. During the early phases, test positivity was very high attributable to the reasons discussed above, i.e., testing was restricted to high-risk symptomatic suspects. However, as the testing criteria became broad and more inclusive, the test positivity trend continued at low levels (<10%) albeit slightly higher at the height of the pandemic, coinciding with the increased levels of transmission.

The proportion of asymptomatic individuals tested could be used to assess the performance of the testing strategies. Certain states like Andhra Pradesh, Gujarat, and Madhya Pradesh continued to test a higher proportion of symptomatic individuals throughout the pandemic, which shows that different states followed different strategies for testing individuals despite ICMR guidelines on testing strategies [28]. The highest proportion of people tested were in Delhi, Goa, Andaman and Nicobar Islands, and Puducherry, all of which were union territories or small states. On the other hand, Madhya Pradesh, Goa, and Nagaland had the highest positivity proportion compared to the other states. Although higher positivity indicates increased transmission, a detailed investigation on why certain states experience exceptionally high positivity is needed focussing on the high under-ascertainment of cases and its implications. Our data analysis captured the massive scale-up of testing facilities across the country in great detail. Indeed, between weeks 27 and 39, testing numbers increased exponentially. One of the essential strategies in the fight against COVID-19 was increased testing and timely and appropriate isolation of positive individuals [29–32]. This strategy was implemented by ramping up the testing to hilt capacity within a record time, albeit not uniformly across the country. The scale-up reduced the time intervals for sample collection and testing and then reporting. The median time for testing after the onset of symptoms was three days in the early phase, but that was reduced to two days in the later phase, which could be attributed to the awareness-raising campaigns that instigated the people for early testing [33].

Each state in India experienced the pandemic differently. Although the federal government coordinated the reporting, the states differed in reporting cases to the laboratory network. In general, almost all the states and UTs were prompt in entering the data. The state of Kerala was notable since it is one of the mainland states that experienced considerable delays in entering positive and negative cases into the database, resulting in an anomalous peak in the national epi-curve at around week 45. Another exception was Telangana, where the number of cases reported became static, with many districts reporting zero cases towards the end of the reference period. Such decline needs scrutiny regarding the factual versus reporting inadequacies. Telangana was the only southern state that reported an incidence of <1 per 100 individuals. Inter-state differences in responding to COVID-19 have been previously explored [34, 35]. Apart from the actual differences in how the pandemic evolved in different geographical regions, other factors that could have contributed to the apparent differences in disease burden include access to testing (especially in the early phases when resources were limited), the extent to which the ICMR testing strategy guidelines were followed and how robustly was the contact tracing carried out. These reasons across states warrant further investigation regarding how much they influenced the incidence and test positivity.

An essential indicator of the efficiency of a nationwide pandemic surveillance system is the timeliness of sample collection after an individual develops symptoms [36, 37]. In our analysis,

we found that in the pandemic's initial phases, the duration between symptom onset and sample collection was three days. In subsequent phases, at least 50% of individuals got tested at symptoms onset, with another 25% delaying their testing by 3 to 4 days. This interval is important because it indicates that the surveillance system became more responsive with time, and individuals became increasingly aware that they needed testing immediately after developing symptoms. A direct effect of this reduction in delay would be more effective isolation of symptomatic patients, early detection of possible contacts, and quarantine. All the above could have played a pivotal role in halting the progress of continued viral transmission [38], preventing infection to severe disease and reducing complications and mortality.

At the national level, the ICMR COVID-19 laboratory surveillance network provided inputs for the national level dashboard for the total number of tests conducted and the number of positives daily. However, no disaggregated information was provided. Each state developed their data analysis methods to steer public health measures of quarantine or containment based on positivity or caseload. It is worth mentioning that the Federal Ministry of Health and Family Welfare operated a system of data collection where each state and union territory reported daily aggregate counts of active cases, recovered individuals and deaths [39].

## Strength and limitations

The key strength of the analysis is the extensive epidemiological profiling of the COVID-19 pandemic based on nearly two hundred million individual records up to the second administrative level in India. We were able to provide insights like the number of symptomatic persons tested, the number of repeat tests, and the evolution and spread of COVID-19 over time and space at the district level for the first time. We provided measures of the timeliness of the surveillance system by depicting the median intervals from symptom onset to sample collection to testing to data entry.

The analysis has few limitations, dictated by the quality of the data available. We did not detect all duplicate entries of persons in the database because that would require retrieving several identifiers, such as the names, residential addresses, zip codes, names of the laboratories/hospitals where the samples were tested and names of the hospitals where the patient was treated (if any). The institutional ethics guidelines restricted the retrieval of these variables. However, we attempted to remove duplicate entries of a test made within the reporting centres based on the date of sample collection to the maximum extent possible. It was not possible to estimate the percentage of records that are potential duplicates of existing individuals. However, the number of such duplicates is unlikely to change our interpretations.

Different kinds of testing kits (different brands of RT-PCR and RAT kits) with varying diagnostic validity were employed to detect the infection, and we were unable to analyse the impact of this on the test positivity. Each of these tests has a different sensitivity and specificity, with sensitivity being higher or equal to specificity. Given this practice, indicators like incidence and test positivity are likely to depend on the proportions of the different tests used. The influence of the test validity and the mix of different tests on such indicators warrant further investigation. We used case-based deletion to deal with the missing information. Less than 1% of the data were missing, and missing data were less prevalent in the later phases of the pandemic. As such, we assume that our analysis was not significantly affected by this issue.

The laboratory surveillance system likely suffered from a certain degree of under-ascertainment of cases, attributed to various reasons. Firstly, testing was voluntary and done only among those who satisfied the eligibility criteria. Since a significant proportion of the COVID-19 infections were likely asymptomatic, this testing strategy may have missed them, and secondly, the varying sensitivity of the testing kits would have led to misclassifying cases into false

negatives. Evidence for this under-ascertainment comes from the infection-to-case ratios (ICR) estimated from the three rounds of nationwide serosurveys [20, 40, 41], which estimates the number of infections undetected in the community for every case identified by the surveillance. The ICR decreased from 81.6 (in May 2020) to 26 (in August 2020) to 27 (in December 2020), suggesting that surveillance improved over time.

## Conclusions

Based on this analysis of ICMR's laboratory surveillance data, we conclude that 1. Incidence of COVID-19 was higher among elderly and men, 2. Positivity and test per capita and % of tests in symptomatic individuals varied in different geographical regions, 3. States within India and districts within the states experienced the pandemic differently at different points of time, 4. The national test positivity stabilized at 6%, with a low proportion of testing among symptomatic individuals, 5. As the surveillance network expanded through the scale-up phases, delays narrowed between the sample collection, testing and reporting and 6. The data management system had issues like duplicate entries and missingness, and the data validity depended on the individual reporting units and state-level authorities.

## Recommendations

Based on our conclusions, we recommend the real-time use of the ICMR's COVID-19 laboratory surveillance database to guide public health measures and testing strategies at state and district levels. Such databases may benefit from having internal validity checks at the time of data collection, entry, and analysis to reduce potential errors. Regions with higher positivity, lower test per capita, and a higher proportion of symptomatic tested should improve test seeking behaviour and access to testing. The timeliness of the surveillance system can be further improved with more robust and targeted communication campaigns, improving access to testing, and considering locally appropriate screening strategies. Ensuring a robust and sensitive surveillance system that can reduce under-ascertainment of cases is essential to control transmission, better understand transmission dynamics, and changes in vaccine effectiveness and in immunity from previous infection and early detection on new variants of concern that may impact the future course of the pandemic. As a long-term strategy, the country needs a plan for establishing data linkages during public health emergencies across databases from laboratories, hospitals, and public health institutions to generate real-time intelligence at various levels on the burden, status of interventions, and their outcomes.

## Supporting information

**S1 Table. Testing, incidence and positivity for COVID-19 among persons by states, gender and age in India (March 2020 to January 2021).**
(DOCX)

**S2 Table. Percentage positive for COVID-19 among persons tested by states, gender and age in India (March 2020 to January 2021).**
(DOCX)

**S3 Table. Testing characteristics of the laboratory surveillance network for COVID-19 by states in India (March 2020 to January 2021).**
(DOCX)

**S4 Table. Overall timeliness of the laboratory surveillance network for COVID-19 by states in India (March 2020 to January 2021).**
(DOCX)

**S5 Table. Timeliness of the laboratory surveillance network for COVID-19 by states and test result in India (March 2020 to January 2021).**
(DOCX)

**S6 Table. Timeliness of the laboratory surveillance network for COVID-19 by states and scale-up phases in India (March 2020 to January 2021).**
(DOCX)

**S1 Fig. Data cleaning work flow for the ICMR COVID-19 laboratory surveillance network in India.**
(PNG)

**S2 Fig. Weekly distribution of positive cases of COVID-19 by total case burden in selected states of India (March 2020 to January 2021).**
(PDF)

**S1 Video. Trend of district-level weekly incidence of COVID-19 cases in India (March 2020 to January 2021).**
(MP4)

**S2 Video. Trend of district-level cumulative incidence of COVID-19 cases in India (March 2020 to January 2021).**
(MP4)

## Author Contributions

**Conceptualization:** Manickam Ponnaiah, Rizwan Suliankatchi Abdulkader, Tarun Bhatnagar, Yogesh Sabde, Manoj Vasanth Murhekar.

**Data curation:** Ramasamy Sabarinathan, Harpreet Singh.

**Formal analysis:** Rizwan Suliankatchi Abdulkader, Jeromie Wesley Vivian Thangaraj, Ramasamy Sabarinathan, Saravanakumar Velusamy, Harpreet Singh.

**Investigation:** Ramasamy Sabarinathan.

**Methodology:** Manickam Ponnaiah, Rizwan Suliankatchi Abdulkader, Tarun Bhatnagar, Jeromie Wesley Vivian Thangaraj, Muthusamy Santhosh Kumar, Ramasamy Sabarinathan, Yogesh Sabde, Manoj Vasanth Murhekar.

**Project administration:** Manickam Ponnaiah, Manoj Vasanth Murhekar.

**Resources:** Harpreet Singh, Manoj Vasanth Murhekar.

**Software:** Rizwan Suliankatchi Abdulkader, Ramasamy Sabarinathan, Saravanakumar Velusamy, Harpreet Singh.

**Supervision:** Manickam Ponnaiah, Manoj Vasanth Murhekar.

**Validation:** Rizwan Suliankatchi Abdulkader.

**Visualization:** Rizwan Suliankatchi Abdulkader.

**Writing – original draft:** Rizwan Suliankatchi Abdulkader.

**Writing – review & editing:** Manickam Ponnaiah, Rizwan Suliankatchi Abdulkader, Tarun Bhatnagar, Jeromie Wesley Vivian Thangaraj, Muthusamy Santhosh Kumar, Ramasamy Sabarinathan, Yogesh Sabde, Harpreet Singh, Manoj Vasanth Murhekar.

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
