## [Decision Letter · Decision Letter 0]

2 Sep 2021

PONE-D-21-24568

Performance of India’s ICMR COVID-19 laboratory surveillance network: Profile of 176 million tested individuals and 188 million tests, March 2020 to January 2021

PLOS ONE

Dear Dr. Murhekar,

Thank you for submitting your manuscript to PLOS ONE. After careful consideration, we feel that it has merit but does not fully meet PLOS ONE’s publication criteria as it currently stands. Therefore, we invite you to submit a revised version of the manuscript that addresses the points raised during the review process.

We look forward to receiving your revised manuscript.

Kind regards,

Sandul Yasobant, PhD

Academic Editor

PLOS ONE

Journal Requirements:

Reviewers' comments:

Reviewer's Responses to Questions

**Comments to the Author**

1. Is the manuscript technically sound, and do the data support the conclusions?

Reviewer #1: Yes

Reviewer #2: Yes

Reviewer #3: Partly

2. Has the statistical analysis been performed appropriately and rigorously? 

Reviewer #1: N/A

Reviewer #2: Yes

Reviewer #3: Yes

3. Have the authors made all data underlying the findings in their manuscript fully available?

Reviewer #1: Yes

Reviewer #2: Yes

Reviewer #3: No

4. Is the manuscript presented in an intelligible fashion and written in standard English?

Reviewer #1: Yes

Reviewer #2: Yes

Reviewer #3: Yes

5. Review Comments to the Author

Reviewer #1: This report is an interesting description of COVID-19 laboratory surveillance with vast amount of data. Very impressive! However, there are some inaccuracies in the data.

Line 125: “testing done on every individual” – this sounds like every individual was tested but this is not the case. Please correct.

The collection of data should be described more in detail. The Methods section starts by referring to the reported data as data source. It would be clearer for the reader if the description of the data should be included here. Furthermore, how was the reported data collected? By a written questionnaire or an interview at the time of the sampling? Who recorded the data, was it a health care professional? How were the data transferred to the database? What symptoms were included as COVID-19 symptoms? What is the “patient category” mentioned on line 160?

On line 161 it is mentioned that the RT-PCR result was entered as three separate results (E, RdsRP and Orf). Was the same method used in every laboratory? How was a positive result defined? Is it enough if one gene target was positive?

On the other hand, on lines 411-412 it is said that “all kinds of testing kits…were employed to detect the infection”. Now this is confusing – did different laboratories use different methods? How is it then possible to record the three above-mentioned genes for each result? Were antigen test results also included in the data?

What were the testing guidelines in the different phases? In the beginning it seems that only symptomatic individuals were tested but later on, over 90% of tested persons were not symptomatic. Why were they then tested?

What does “without result” mean? Does it mean the result was not recorded or the test was inconclusive or the test tube had to be discarded for some reason, pre-analytical or analytical?

If I understand correctly, the same person could be included in the numbers several times. Do the authors have any idea how much of these repeated testings are included in the numbers?

Could the age group 18-60 be divided in smaller groups? In many previous studies the proportion of females among the tested has been higher than males. It has been suggested that females take often better care of their health and seek testing more eagerly than males. This study has a different result which should be discussed.

The language needs checking (for example but not limited to: atleast > at least; infact > in fact).

Reviewer #2: The article presents a thorough analysis of the Indian approach to COVID-19 testing, providing comparisons of both testing capacity and results among the various Indian states. The statistical analyses are sound and the presentation of the data intelligent and straightforward. The use of GIS figures is particularly handy for guiding the reader through the differences between states with regard to testing and results. The manuscript requires some linguistic editing, most of which I have done in the attached MS Word file.

Reviewer #3: Very relevant work and insights for understanding surveillance of COVID-19 across the globe and a good effort to upgrade surveillance in India. Findings should inform need to continue upgrading surveillance of COVID-19 , including variants and vaccine effectiveness.

Comments are presented by section:

Title:

Performance is a broad term. Consider Testing, timeliness and positivity.

Abstract:

“ Durantion between the various surveillance activities was acceptable indicating a good responsiveness

of the surveillance system” Consider Timeliness of the surveillance system, instead of duration.

Even tough the scaling up was impressive for such a large country, a brief reference to potential under-ascertainment of infections could be refered in the abstract(need for further evaluation to incresase surveillance system sensitivity.

Introduction

The testing strategy in the country must be described in Introduction and refered in Discussion. (what symptoms were tested, criteria to test contacts of cases and other testing strategies in specific settings that may exist)

A brief reference to testing accessibility by the population is important to put the surveillance system into context. Is there a syndromic surveillance system or a national health line to report symptoms and prescribe tests?

What was necessary for somenone to be able to be tested?

Were all test PCR tests?

Methods

“All the reporting units in the network entered the data on a daily basis from all the 154

districts of the 37 states and union territories of India.” Manual entry or automatic processes?Updates were made manually?

“Due to the large size of the data, all calculations were performed in a server” – What server. The readers may not be familiar. Could be clarified.

Results

Man had higher testing incidence than woman. This is different than other countries (for example in Europe and could be briefly approached in discussion .

% od individuals that were symptomatic when tested seems extremely low considering other countries experience. Should be discussed.

Positiviy should be described in different time periods. Epicurves of positivity could be shown as a measure of potential under-ascertainment in different time periods

Timeliness should be discussed in terms of possible improvements specially from symptom onset to testing as this has very relevant public health implications.

Assymetry in testing in different regions and difference in Percentage of persons positive among tested could be briefly discussed further discussed considering accessibility to testing, and demographic factors and related to different surveillance sensitivity in different areas.

“Spatial distribution of incidence was similar to the previous with the 222 striking exception of Maharashtra which showed high incidence and positivity (despite 223 moderate testing) and Telangana which showed low incidence and positivity (despite higher 224 testing)”

This should be discussed in the discussion chapter considering possibly adequate levels of testing in places lije Telangana and suboptimal testing in places like Maharashtra were higher under-ascertainment of infection is likely. Because of this Telangana showed low incidence and positivity possibly also because of higher testing and not only despite of. Consider changing the word despite because it implies that this was not expected . Places with higher testing capacity tend to have lower positivity rates and lower incidences if testing and contact tracing works well( but are influenced by demography).

Part of the results description that are obvious from figures and tables and do not imply any specific comments in th Discussion could be omitted to make more space for discussion of the more important findings and interpretation.

Discussion

“The higher incidence 327

among elderly was probably due to their aging immunological and physiological status and 328

that of presence of co-morbidities such as diabetes and hypertension”

Because they have more severe symptoms and may be more interested in testing. Incidence in serologic studies is usually not higher in older people. What usually happens is less infections are detected in younger because of milder symptoms and less motivation or access to testing. This should be refered.

Higher incidence in man must refer also possible higher test seeking behavior in men in India´s cultural context.´

Changes in testing criteria must be presented in Introduction and discussed as relevant in discussion.

“As the pandemic progressed, the testing of asymptomatic individuals could reflect the 353

performance of the testing strategies. Certain states like Andhra Pradesh, Gujarat and 354

Madhya Pradesh still continued to test a higher proportion of symptomatic individuals despite 355

this fact .This shows that states followed different strategies for testing individuals despite 356

regular revisions of ICMR testing strategies.” - Or that testing should be scaled up in this regions to avoid higher under-ascertainment of cases.

“Although higher positivity indicates increased transmission, a detailed investigation on 361

why certain states experience exceptionally high positivity needs further investigation” Refer that this may imply high levels of under-ascertainment, undetected cases.

Avoid repeating results in discussion. Do it only briefly to discuss relevant aspects.

“One of the most important strategies in the fight 366

against COVID-19 was more testing and more isolation” – could you clarify “more isolation”

“lthough, all kinds of testing kits (like Rapid Antigen Tests and different brands of RT-PCR) 411

with varying diagnostic validity were employed to detect the infection, we were unable to 412

analyse the impact of this on the test positivity”

Why was it not possible? Was “Test method” not registered in the database?

In the end of strengths and limitations it is of high importance to refer surveillance system sensitivity , potential high under-ascertainment, specially n regions with higher positivity and lower tests per capita. This should inform needs to uprgrade surveillance.

Include references related to under-ascertainment/under-detection of cases/infection.

It is important to discuss, based on relevant references, serologic studies and others what was the level of under-ascertainment/under-detection of cases/infection during the whole period and in different periods.

Conclusions

Include “positivity and test per capita and % test in symptomatic individual varied in different regions.

Reccomendations

Consider including “ Regions with higher positivity,lower test per capita and higher % of symptomatic tested should improve test seeking behavior and access to testing as they have higher under-asdcertainemnte/under-detection of cases/infection.

Timeliness can be further improved with stronger and targeted communication campaigns and improving/facilitating access to testing and considering other broader screening strategies.

Can be of relevance to consider a brief note in reccomendations or discussion to the importance of guaranteeing high surveillance system sensitivity/ case ascertainment case detection to allow for control of transmission, understanding transmission dynamics and changes in vaccine effectiveness and early detection and research on new variants of concern that may impact the future. This is why surveillance improvement in COVID-19 is so relevant everywhere for the world.

6. PLOS authors have the option to publish the peer review history of their article (what does this mean?). If published, this will include your full peer review and any attached files.

Reviewer #1: No

Reviewer #2: **Yes: **Nicholas Saadah

Reviewer #3: No

---

## [Author Response · Author response to Decision Letter 0]

9 Sep 2021

Responses to Reviewer's comments

Reviewer #1:

This report is an interesting description of COVID-19 laboratory surveillance with vast amount of data. Very impressive! However, there are some inaccuracies in the data.

Response: We thank the reviewer for this comment. We have addressed the comments as follows.

Query 1: Line 125: “testing done on every individual” – this sounds like every individual was tested but this is not the case. Please correct.

Response: We thank the reviewer for pointing out this error. The sentence has been modified as follows. [Line 70] 

“This network maintains data on individuals who have been tested for COVID-19 in ICMR-approved laboratories in the country.”

Query 2: The collection of data should be described more in detail. The Methods section starts by referring to the reported data as data source. It would be clearer for the reader if the description of the data should be included here. Furthermore, how was the reported data collected? By a written questionnaire or an interview at the time of the sampling? Who recorded the data, was it a health care professional? How were the data transferred to the database? What symptoms were included as COVID-19 symptoms? What is the “patient category” mentioned on line 160?

Response: The relevant details have been added as follows in the ‘Data source’ subheading of the ‘Methods’ section. [Line 105-135]

“Data source

The reported data of the individuals tested at the laboratories and testing centres accredited to the ICMR COVID-19 diagnosis and surveillance network. This data was collected through a mobile application developed for this purpose by ICMR. The form used to collect data was called the “ICMR Specimen Referral Form for COVID-19 (SARS-CoV2)”.[10] The laboratory personnel interviewed the individual at the time of sample collection using a print version of the form. Information was collected on name, age, sex, address, contact number, whether follow up sample, type of specimen (nasal, throat, nasopharyngeal or others), type of patient category (whether it was for routine surveillance in containment/non-containment zones, Severe Acute Respiratory Illness/Influence Like Illness patients in hospitals, high-risk asymptomatic/symptomatic contacts, pregnant women, symptomatic neonates or patient-initiated situations like travel), clinical signs and symptoms (like cough, breathlessness, sore throat, diarrhoea, nausea, chest pain, vomiting, haemoptysis, nasal discharge, fever, body ache, sputum, abdominal pain), co-morbidities (like diabetes, hypertension, malignancy, chronic lung, kidney or liver diseases or immunocompromised conditions), and hospitalisation details (if applicable). Each reporting centre/laboratory manually entered the above information in the mobile/web application on the day of sample collection. All laboratories in the network used one of the ICMR approved Reverse Transcriptase – Polymerase Chain Reaction (RT-PCR) kits or Rapid Antigen Test (RAT) kits for detection SARS-CoV-2 infection. When RT-PCR kits were used for testing, the laboratories were required enter the results of the individual genes and an overall positive or negative result. The algorithms for declaring an RT-PCR test as positive were dependent on type of kit used. Early in the pandemic when only one approved kit was available, the testing happened in two stages – if E gene was not detected, the test was declared negative. If E gene was detected, ORF and RdRp genes were tested and result was given as positive if one of them was detected and as negative otherwise. Nowadays, all the kits are multiplex assays where are all three genes are tested together and a positive result is declared if two out of three genes are detected with a cycle threshold value of <35 and as negative otherwise. When RAT kits were used, only an overall positive or negative result was entered. Each record was updated with the test outcome whenever the results became available.”

Query 3: On line 161 it is mentioned that the RT-PCR result was entered as three separate results (E, RdRP and Orf). Was the same method used in every laboratory? How was a positive result defined? Is it enough if one gene target was positive? On the other hand, on lines 411-412 it is said that “all kinds of testing kits…were employed to detect the infection”. Now this is confusing – did different laboratories use different methods? How is it then possible to record the three above-mentioned genes for each result? Were antigen test results also included in the data?

Response: To provide more clarity, this information has been updated in the ‘Data source’ subheading of the ‘Methods’ section. [Line 105-135]

“Data source

The reported data of the individuals tested at the laboratories and testing centres accredited to the ICMR COVID-19 diagnosis and surveillance network. This data was collected through a mobile application developed for this purpose by ICMR. The form used to collect data was called the “ICMR Specimen Referral Form for COVID-19 (SARS-CoV2)”.[10] The laboratory personnel interviewed the individual at the time of sample collection using a print version of the form. Information was collected on name, age, sex, address, contact number, whether follow up sample, type of specimen (nasal, throat, nasopharyngeal or others), type of patient category (whether it was for routine surveillance in containment/non-containment zones, Severe Acute Respiratory Illness/Influence Like Illness patients in hospitals, high-risk asymptomatic/symptomatic contacts, pregnant women, symptomatic neonates or patient-initiated situations like travel), clinical signs and symptoms (like cough, breathlessness, sore throat, diarrhoea, nausea, chest pain, vomiting, haemoptysis, nasal discharge, fever, body ache, sputum, abdominal pain), co-morbidities (like diabetes, hypertension, malignancy, chronic lung, kidney or liver diseases or immunocompromised conditions), and hospitalisation details (if applicable). Each reporting centre/laboratory manually entered the above information in the mobile/web application on the day of sample collection. All laboratories in the network used one of the ICMR approved Reverse Transcriptase – Polymerase Chain Reaction (RT-PCR) kits or Rapid Antigen Test (RAT) kits for detection SARS-CoV-2 infection. When RT-PCR kits were used for testing, the laboratories were required enter the results of the individual genes and an overall positive or negative result. The algorithms for declaring an RT-PCR test as positive were dependent on type of kit used. Early in the pandemic when only one approved kit was available, the testing happened in two stages – if E gene was not detected, the test was declared negative. If E gene was detected, ORF and RdRp genes were tested and result was given as positive if one of them was detected and as negative otherwise. Nowadays, all the kits are multiplex assays where are all three genes are tested together and a positive result is declared if two out of three genes are detected with a cycle threshold value of <35 and as negative otherwise. When RAT kits were used, only an overall positive or negative result was entered. Each record was updated with the test outcome whenever the results became available.”

Query 4: What were the testing guidelines in the different phases? In the beginning it seems that only symptomatic individuals were tested but later on, over 90% of tested persons were not symptomatic. Why were they then tested?

Response: The testing guidelines evolved throughout the reference period. Initially, only symptomatic individuals, high-risk contacts and travellers from affected countries were tested. We have elaborated the change testing guidelines in the introduction and discussion section as follows.

Introduction [Line 73—88]

“India’s COVID-19 testing strategy evolved with the changing needs of the pandemic. In March 2020, when the testing resources were limited, only high-risk symptomatic individuals, international inbound travellers, high-risk contacts and patients with Severe Acute Respiratory Illness (SARI) were eligible for testing. During the later phases of the pandemic, the eligibility criteria was expanded to include, asymptomatic contacts, surveillance of symptomatic persons in containments zones, patients with Influenza Like Illness (ILI) and, migrant workers. More recently, patient-initiated testing was also added on where situations such as international travel, screening before undergoing medical/surgical procedures were also eligible for testing. A number of public-centric initiatives were taken by the government to provide free and wide access to testing. Persons who wanted testing were asked to contact the nearest public health care facility where testing was provided free of cost. Testing was carried out at homes of symptomatic patients/contacts of laboratory confirmed cases by healthcare workers during routine house-to-house fever surveillance. Mobile vans located at strategic places in major towns and cities also provided testing facilities to people who wanted to get tested. Apart from these government initiatives, many private labs and hospitals also provided fee-based testing.[2,3]”

Discussion [Line 380-393]

“Periodic changes in the testing strategy at different phases of the pandemic were reflected in the upward trends in the frequency of asymptomatic individuals and the declining trends in repeat testing.[28] Initially, when the testing strategy was exclusively done among high-risk category, the percentage of symptomatic individuals tested was high. However, later on when the criteria were made more inclusive, the same indicator tapered off to around 7-10% Changes in testing criteria were also reflected in the percentage of repeat testing of the individuals. During the early phases, tests were repeated every week or so for hospitalized patients and a negative test was mandatory before discharge. When this criterion was relaxed the repeat testing proportions tapered to around 5%. The time trend in test positivity reflected the increased levels of transmission. During the early phases, test positivity was very high attributable to the reasons discussed above, i.e., testing was restricted to high-risk symptomatic suspects. However, as the testing criteria became broad and more inclusive, the test positivity trend continued at low levels (<10%) albeit slightly higher at the height of the pandemic, coinciding with the increased levels of transmission.”

Query 5: What does “without result” mean? Does it mean the result was not recorded or the test was inconclusive or the test tube had to be discarded for some reason, pre-analytical or analytical?

Response: It means the outcome of the test was not entered in the database by the laboratory personnel. The reason for this missing data was unknown.

Query 6: If I understand correctly, the same person could be included in the numbers several times. Do the authors have any idea how much of these repeated testing are included in the numbers?

Response: Table 1, which gives the results of individuals does not include repeated testing. However, table 2 and table 3 which are based number of tests include repeated testing on the same person. The percentage of repeat testing is 5.1%.

Query 7: Could the age group 18-60 be divided in smaller groups? In many previous studies the proportion of females among the tested has been higher than males. It has been suggested that females take often better care of their health and seek testing more eagerly than males. This study has a different result which should be discussed.

Response: In India, the proportion of females among tested is lower than the males. Even if the age is further categorized into smaller groups, the finding remains the same. This has been discussed further in the third paragraph of discussion as follows.[Line 359-365]

“Further, it was seen that the testing rate was higher in men, than in women across all states, in contrast to developed countries in the west.[22] Among those who were tested, the proportion of women was highest in Andhra Pradesh (46.4%) and lowest in Dadra and Nagar Haveli (23.4%). It has been previously shown that young women in certain states of India were less likely than men to practice key COVID-19 preventive behaviours.[23] Thus, such low frequency of testing behaviour among women was reflected in low prevalence among women as well.”

Query 8: The language needs checking (for example but not limited to: atleast > at least; infact > in fact).

Response: We have edited the language as per the reviewer’s suggestions.

Reviewer #2: 

The article presents a thorough analysis of the Indian approach to COVID-19 testing, providing comparisons of both testing capacity and results among the various Indian states. The statistical analyses are sound and the presentation of the data intelligent and straightforward. The use of GIS figures is particularly handy for guiding the reader through the differences between states with regard to testing and results. The manuscript requires some linguistic editing, most of which I have done in the attached MS Word file.

Response: We thank the reviewer for the constructive feedback and editing. We have accepted all the edits made and also, made language edits wherever required.

Query 1: It is not clear to me what is meant by ‘at the individual level’ here as incidence is a population level measurement. Consider removing this phrase.

Response: We agree with the reviewer and have removed the phrase ‘at the individual level.’ [Line 203]

Query 2: How about rephrasing this as ‘6.1% of individuals tested returned a positive result’ or something similar – that’s a bit clearer than what is currently written.

Response: The revised sentence now reads “6.1% of individuals tested returned a positive result.” [Line 33]

Query 3: Don’t report an approximate number here – report the actual percentage.

Response: The exact percentage was 75.6% and this has been mentioned now in the text. [Line 211]

Query 4: It is not clear to me what you mean by this word – I wonder if it might be better to say ‘In bringing the COVID-19 testing strategy to operation…’

Response: The sentence now reads as below.[Line-288]

“In bringing the COVID-19 testing strategy to operation, the Indian Ministry of Health and Family Welfare established a network of COVID-19 testing laboratories under the auspices of the national medical research body, the Indian Council of Medical Research (ICMR).”

Query 5: Please specify what this is the incidence of – positive COVID tests, yes?

Response: Yes, it is the incidence of COVID-19 among individuals based on positive test. For clarity, we have changed the sentence to [Line 208]. 

“The overall incidence of COVID-19 was 0.8% and 12,584 persons per 100,000 population were tested.”

Query 6: Why ‘about’ – is this not an exact number?

Response: 6.1% is the exact number. The sentence now reads as follows.[Line 201] 

“6.1% of the persons tested were positive for SARS-CoV-2, with the highest positivity (10.9%) among those aged above 60 years.(Table 1)”

Query 7: Of what? Please specify.

Response: It refers to the incidence of people who are COVID-19 positive in the population. The sentence now reads as follows.[Line 2019] 

“The highest incidence of COVID-19 in the population was seen in Goa (3.7%) and Delhi (3.3%) and the highest test positivity was seen in Maharashtra (15.5%) and Goa (13%).”

Query 8: Why are these results not presented either in the article or in the supplement?

Response: This web application is meant to provide in-depth details of the analyses conducted in the paper for readers who are interested at district level estimates. State-level results shown in this web application have been discussed throughout the paper and the same information has been reproduced in the supplement. For example, S1-3 Tables contain the same data as given in the summary indicators of the web application. However, district-level results were not discussed in the main paper or in the supplement.

Query 9: Please provide the percentage increase

Response: The sentence has been reframed as follows.[Lin2 236] 

“Kerala reported the highest weekly increase (by 271 times) in terms of absolute case count between ISO weeks 51 and 52…”

Query 10: This sentence is confusing to me – could you please re-write to make more clear? I believe you are saying that Telangana was an outlier in that it reported fewer cases as the pandemic progressed? If so, perhaps something like: Telangana was an outlier compared to other states as it reported fewer cases with time with half of its districts reporting no cases towards the end of the observation period.

Response: The sentences have been rephrased as follows. [Line 251] 

“Telangana was an outlier as compared to other southern states as it reported fewer cases with time with half of its districts reporting no cases towards the end of the observation period.(Figure 4II.B)”

Query 11: You state you will present median (IQR), but you present median (range). Please make consistent.

Response: We thank the reviewer for point out this error. [Line 296]

Range has been changed to IQR.

Query 12: Please list the primary outcome as well as the IQRs here.

Response: We have added all the primary outcomes about the timelines. The sentence now reads as follows.[Line 296] 

“Median (IQR) duration between symptom onset and collection was 2 (IQR: 0 to 3) days and that between sample collection and testing, and between testing and reporting were 0 (IQR: 0 to 1) days.”

Query 12: I believe with ‘distributions’ you are referencing the variance in the measurements (i.e. how narrow the distributions were). Better to present this as 95% confidence intervals (as these are the standard way of presenting the width of a distribution).

Response: We thank the reviewer for this observation. We did not intend to refer to the 95% confidence interval. The point we wanted to make is that the IQR became narrow in the later phases of the scale-up as compared to the earlier phases as seen in figure 7. III. 

We have revised the sentence which now reads.[Line 300]

“Across the scale-up phases, the symptom to collection duration decreased from 3 days to 2 days and the other intervals decreased from 1 day to zero day with narrower inter-quartile ranges.”

Query 13: Please present measures of variance (e.g. 95% confidence intervals, IQRs, or ranges) for these values. It is not clear to me what you mean by ‘across test results’ here – please re-word.

Response: We have now provided the IQR for the indicators mentioned. The modified paragraph reads as follows.[Line 311-320]

“Among the states, symptom to collection duration varied from zero to 3.5 days and collection to testing duration was typically less than one day. In contrast, the testing to entry time was longer for Andaman and Nicobar Islands (13 days, IQR: 2 to 26), Kerala (10 days, IQR: 2 to 22) and Nagaland (4 days, IQR: 1 to 12). In general, union territories and north-eastern and island states experienced longer durations than those of the large states for positive test results and across all scale-up phases. Positive tests were generally entered more or as quickly as negative tests with the exception of Nagaland and Telangana. Nagaland took the longest time (7 days, IQR: 3 to 10) to enter positive tests followed by Kerala (5 days, IQR: 1 to 15) and Andaman and Nicobar Islands (4 days, IQR: 1 to 21 days).(S4-6 Tables)”

Query 13: It is not clear to me what this means. Please expand.

Response: In this paragraph we wanted to say that identification of all duplicates would have required us to retrieve several identifying variables as mentioned below. This was not permitted by the ethics committee that approved the proposal. Therefore, we did not remove duplicates based on these identifying variables. For the sake of clarity, the modified paragraph now reads as follows: [Line 467-474]

“We did not detect all duplicate entries of persons in the database because that would require retrieving several identifiers, such as name of the individual, residential address, zip codes, name of the laboratory/hospital where the sample was tested and name of the hospital where the patient was treated (if any). Retrieval of these variables were restricted by the institutional ethics guidelines. However, we attempted to remove duplicate entries of a test made within the reporting centres based on the date of sample collection to the maximum extent possible.”

Query 14: This belongs in results, not in the discussion.

Response: This finding has been shifted to the results section as a separate paragraph as shown below: [Line 322-325]

“Missingness of data

A database of this size is expected to have missing data. We found that across the variables used in the analysis, missingness ranged from 0.01% for age of the individual to 0.7% for outcome of the test result. Date of testing was missing in 0.08% of the entries.”

Query 15: You can either write ‘at state and district levels’ of ‘at the state or district level’ – both are correct and are equivalent statements.

Response: The modified sentence reads as follows. [Line 516]

“On the basis of our conclusions, we recommend real-time use of the ICMR COVID-19 laboratory surveillance database to guide public health measures and testing strategies at state and district levels.”

Query 16: This sentence lacks a verb – I think maybe you mean to write. ‘As a long-term strategy, a plan for establishing data linkages during public health emergencies across databases from laboratory, hospitals, and public health authorities IS NEEDED in order to generate real-time intelligence at various levels on the burden, status of interventions, and their outcomes.’ Is this correct? If not, please add the appropriate verb to make this sentence complete.

Response: We have accepted the reviewer’s suggestion and the modified sentence as follows. [Line 528] 

“As a long-term strategy, a plan for establishing data linkages during public health emergencies across databases from laboratory, hospitals, and public health authorities is needed in order to generate real-time intelligence at various levels on the burden, status of interventions, and their outcomes.”

Reviewer #3: 

Very relevant work and insights for understanding surveillance of COVID-19 across the globe and a good effort to upgrade surveillance in India. Findings should inform need to continue upgrading surveillance of COVID-19 , including variants and vaccine effectiveness.

Comments are presented by section:

Query 1: Title: Performance is a broad term. Consider Testing, timeliness and positivity.

Response: The title has been changed. [Line 1]

“COVID-19 testing, timeliness and positivity from ICMR’s laboratory surveillance network in India: Profile of 176 million individuals tested and 188 million tests, March 2020 to January 2021”

Query 2: Abstract: “Duration between the various surveillance activities was acceptable indicating a good responsiveness of the surveillance system” Consider Timeliness of the surveillance system, instead of duration.

Response: The sentence has been changed to “Timeliness between the various surveillance activities was acceptable indicating a good responsiveness of the surveillance system.” [Line 43]

Query 3: Even though the scaling up was impressive for such a large country, a brief reference to potential under-ascertainment of infections could be referred in the abstract (need for further evaluation to increase surveillance system sensitivity.

Response: We agree that there could be a potential for under-ascertainment of infections considering the evolving testing guidelines and expansion of laboratories across the country. However, it is beyond the scope of the present analysis to detect the level of under-ascertainment. Nevertheless, we have added a paragraph in the discussion section referring to this issue. [Line 491-501] 

“It is likely that the laboratory surveillance system suffered from a certain degree of under-ascertainment of cases. This could be attributed to various reasons. Firstly, testing was voluntary and done only among those who satisfied the eligibility criteria. Since >90% of the infections were asymptomatic, this testing strategy was likely to have missed them and secondly, the sensitivity of the testing kits ranged from 70 to 90%, which again would have led to misclassifying cases into false negatives. Evidence for this under-ascertainment comes from the infection-to-case ratios (ICR) estimated over time from the three rounds of nation-wide sero-surveys in India [20,41,42], which estimates the number likely infections undetected in the community for every case identified by the surveillance. However, the ICR decreased from 81.6 (in May, 2020) to 26 (in August, 2020) to 27 (in December, 2020) suggesting that the sensitivity of the surveillance improved over time.”

Query 4: Introduction: The testing strategy in the country must be described in Introduction and referred in Discussion. (what symptoms were tested, criteria to test contacts of cases and other testing strategies in specific settings that may exist) A brief reference to testing accessibility by the population is important to put the surveillance system into context. Is there a syndromic surveillance system or a national health line to report symptoms and prescribe tests? What was necessary for someone to be able to be tested?

Response: The following paragraph has been added in the introduction and also referred to in the discussion. [Line 73-88]

India’s COVID-19 testing strategy evolved with the changing needs of the pandemic. In March 2020, when the testing resources were limited, only high-risk symptomatic individuals, international inbound travellers, high-risk contacts and patients with Severe Acute Respiratory Illness (SARI) were eligible for testing. During the later phases of the pandemic, the eligibility criteria was expanded to include, asymptomatic contacts, surveillance of symptomatic persons in containments zones, patients with Influenza Like Illness (ILI) and, migrant workers. More recently, patient-initiated testing was also added on where situations such as international travel, screening before undergoing medical/surgical procedures were also eligible for testing. A number of public-centric initiatives were taken by the government to provide free and wide access to testing. Persons who wanted testing were asked to contact the nearest public health care facility where testing was provided free of cost. Testing was carried out at homes of symptomatic patients/contacts of laboratory confirmed cases by healthcare workers during routine house-to-house fever surveillance. Mobile vans located at strategic places in major towns and cities also provided testing facilities to people who wanted to get tested. Apart from these government initiatives, many private labs and hospitals also provided fee-based testing.[2,3]

The following paragraph has been added in the discussion section: [Line 380-393]

“Periodic changes in the testing strategy at different phases of the pandemic were reflected in the upward trends in the frequency of asymptomatic individuals and the declining trends in repeat testing.[28] Initially, when the testing strategy was exclusively done among high-risk category, the percentage of symptomatic individuals tested was high. However, later on when the criteria were made more inclusive, the same indicator tapered off to around 7-10% Changes in testing criteria were also reflected in the percentage of repeat testing of the individuals. During the early phases, tests were repeated every week or so for hospitalized patients and a negative test was mandatory before discharge. When this criterion was relaxed the repeat testing proportions tapered to around 5%. The time trend in test positivity reflected the increased levels of transmission. During the early phases, test positivity was very high attributable to the reasons discussed above, i.e., testing was restricted to high-risk symptomatic suspects. However, as the testing criteria became broad and more inclusive, the test positivity trend continued at low levels (<10%) albeit slightly higher at the height of the pandemic, coinciding with the increased levels of transmission."

Query 5: Were all test PCR tests?

Response: The tests used in this surveillance system were RT-PCR and RAT kits. This has been added now in the data source section. “All laboratories in the network used one of the ICMR approved Reverse Transcriptase – Polymerase Chain Reaction (RT-PCR) kits or Rapid Antigen Test (RAT) kits for detection SARS-CoV-2 infection.” (Line 122)

Query 6: Methods: 

“All the reporting units in the network entered the data on a daily basis from all the 154

districts of the 37 states and union territories of India. Manual entry or automatic processes? Updates were made manually? “Due to the large size of the data, all calculations were performed in a server” – What server. The readers may not be familiar. Could be clarified.

Response: The details of the data entry process are now added in the data source section for clarity. The initial data entry was performed manually by the laboratory personnel and also, updated manually after the test results were available.

We have removed the line “Due to the large size of the data, all calculations were performed in a server.”

Query 7: Results: Man had higher testing incidence than woman. This is different than other countries (for example in Europe and could be briefly approached in discussion.

Response: This has been discussed in the third paragraph of the discussion section as follows.[Line 737-743]

“Further, it was seen that the testing rate was higher in men, than in women across all states, in contrast to developed countries in the west.[22] Among those who were tested, the proportion of women was highest in Andhra Pradesh (46.4%) and lowest in Dadra and Nagar Haveli (23.4%). It has been previously shown that young women in certain states of India were less likely than men to practice key COVID-19 preventive behaviours.[23] Thus, such low frequency of testing behaviour among women was reflected in low prevalence among women as well.”

Query 8: % of individuals that were symptomatic when tested seems extremely low considering other countries experience. Should be discussed. ?

Response: The following lines have been added in the discussion section.[Line 713-718]

“The overall percentage of symptomatic individual who were tested was lower in our study. This is due to two main reasons – firstly, the testing criteria was kept relatively broad and a large proportion of testing was carried out among asymptomatic individuals who were contacts of cases or picked up for random testing in containment areas [14,15] and secondly, data on symptoms may have been inaccurately reported, collected, or entered in the database.”

Query 9: Positivity should be described in different time periods. Epicurves of positivity could be shown as a measure of potential under-ascertainment in different time periods.

Response: The time trend of positivity was depicted in figure 6. And the following lines have been added in the discussion section now.[Line 773-778]

“The time trend in test positivity reflected the increased levels of transmission. During the early phases, test positivity was very high attributable to the reasons discussed above, i.e., testing was restricted to high-risk symptomatic suspects. However, as the testing criteria became broad and more inclusive, the test positivity trend continued at low levels (<10%) albeit slightly higher at the height of the pandemic, coinciding with the increased levels of transmission.”

Query 10: Timeliness should be discussed in terms of possible improvements specially from symptom onset to testing as this has very relevant public health implications.

Response: The following paragraph has been added in the discussion. [Line 845-867]

“An important indicator of the efficiency of a nationwide pandemic surveillance system is the timeliness of sample collection after an individual develops symptoms.[37,38] In our analysis, we found that in the initial phases of the pandemic the duration between symptom onset and sample collection was three days. In the subsequent phases, we found that the median duration was reduced to zero. This is an important finding because it means that the surveillance system became more responsive and individuals were becoming increasingly aware of the need to get tested immediately after they develop symptoms. An direct effect of this reduction in delay would be more effective isolation of symptomatic patients, early detection of possible contacts and their quarantine. This could have played an pivotal role in halting the progress of continued viral transmission [39], preventing the progress of infection to severe disease and reduction of complications and mortality.”

Query 11. Asymmetry in testing in different regions and difference in Percentage of persons positive among tested could be briefly discussed further discussed considering accessibility to testing, and demographic factors and related to different surveillance sensitivity in different areas.

Response: The following paragraph has been added in the discussion.[Line 836-843]

“Inter-state differences in the response to COVID-19 in India have been previously explored.[35,36] Apart from the actual differences in the how the pandemic evolved in different geographical regions, other factors that could have contributed to the apparent differences in disease burden include access to testing (especially in the early phases when resources were limited), the extent to which the ICMR testing strategy guidelines were followed and how robustly was the contact tracing carried out. These reasons across states warrant further investigation in terms of how much they influenced the incidence and test positivity.”

Query 12: “Spatial distribution of incidence was similar to the previous with the striking exception of Maharashtra which showed high incidence and positivity (despite moderate testing) and Telangana which showed low incidence and positivity (despite higher testing)”

This should be discussed in the discussion chapter considering possibly adequate levels of testing in places lije Telangana and suboptimal testing in places like Maharashtra were higher under-ascertainment of infection is likely. Because of this Telangana showed low incidence and positivity possibly also because of higher testing and not only despite of. Consider changing the word despite because it implies that this was not expected . Places with higher testing capacity tend to have lower positivity rates and lower incidences if testing and contact tracing works well( but are influenced by demography).

Response: The word despite has been replaced. The sentence now reads as follows.[Line 400] 

“Spatial distribution of incidence was similar to that of testing with the striking exception of Maharashtra which showed high incidence and positivity (where testing was moderate) and Telangana which showed low incidence and positivity (where testing was high).”

The relationship between testing levels and positivity and incidence has been discussed in the discussion section as follows: [Line 750-763]

“The relationship between the level of testing and positivity and incidence has been established previous studies.[26,27] When the testing levels are relatively high, the incidence and test positivity are low and vice versa, assuming a constant transmission. In our study as well, this relationship was noted with a few exceptions as noted in the case of Telangana and Maharashtra. However, other explanations including the possibility high levels of under-ascertainment of cases could also be operating within the states.”

Query 13: Part of the results description that are obvious from figures and tables and do not imply any specific comments in the Discussion could be omitted to make more space for discussion of the more important findings and interpretation.

Response: We have removed repetition of results in the discussion section as suggested by the reviewer.

Query 14: Discussion “The higher incidence among elderly was probably due to their aging immunological and physiological status and that of presence of co-morbidities such as diabetes and hypertension” Because they have more severe symptoms and may be more interested in testing. Incidence in serologic studies is usually not higher in older people. What usually happens is less infections are detected in younger because of milder symptoms and less motivation or access to testing. This should be referred.

Response: We thank the reviewer for this insightful comment and we agree with it. The following sentence has been added in the relevant discussion section. [Line 730]

“It could be also explained by the health seeking behaviour of older adults compared to young people. Young people tend to have milder symptoms and to have less interest in getting tested. This can be seen in serologic studies where the sero-positivity is comparable between older and younger adults.[20]”

Query 15: Higher incidence in man must refer also possible higher test seeking behavior in men in India´s cultural context.´

Response: The reviewer is correct in pointing this out. We have added this in the discussion section as follows.[Line 733]

“Similarly, the higher incidence in males could be attributable to biological (such as higher expression of angiotensin-converting enzyme-2 receptors), sociological (such as poorer lifestyle choices and riskier behaviours) factors and cultural factors (such as higher access to testing facilities and importance given to men when it comes to health-seeking) in the Indian context.[21]”

Query 16: Changes in testing criteria must be presented in Introduction and discussed as relevant in discussion.

Response: The testing guidelines evolved throughout the reference period. Initially, only symptomatic individuals, high-risk contacts and travellers from affected countries were tested. We have elaborated the change testing guidelines in the introduction and discussion section as follows.

Introduction [Line 73—88]

“India’s COVID-19 testing strategy evolved with the changing needs of the pandemic. In March 2020, when the testing resources were limited, only high-risk symptomatic individuals, international inbound travellers, high-risk contacts and patients with Severe Acute Respiratory Illness (SARI) were eligible for testing. During the later phases of the pandemic, the eligibility criteria was expanded to include, asymptomatic contacts, surveillance of symptomatic persons in containments zones, patients with Influenza Like Illness (ILI) and, migrant workers. More recently, patient-initiated testing was also added on where situations such as international travel, screening before undergoing medical/surgical procedures were also eligible for testing. A number of public-centric initiatives were taken by the government to provide free and wide access to testing. Persons who wanted testing were asked to contact the nearest public health care facility where testing was provided free of cost. Testing was carried out at homes of symptomatic patients/contacts of laboratory confirmed cases by healthcare workers during routine house-to-house fever surveillance. Mobile vans located at strategic places in major towns and cities also provided testing facilities to people who wanted to get tested. Apart from these government initiatives, many private labs and hospitals also provided fee-based testing.[2,3]”

Discussion [Line 380-393]

“Periodic changes in the testing strategy at different phases of the pandemic were reflected in the upward trends in the frequency of asymptomatic individuals and the declining trends in repeat testing.[28] Initially, when the testing strategy was exclusively done among high-risk category, the percentage of symptomatic individuals tested was high. However, later on when the criteria were made more inclusive, the same indicator tapered off to around 7-10% Changes in testing criteria were also reflected in the percentage of repeat testing of the individuals. During the early phases, tests were repeated every week or so for hospitalized patients and a negative test was mandatory before discharge. When this criterion was relaxed the repeat testing proportions tapered to around 5%. The time trend in test positivity reflected the increased levels of transmission. During the early phases, test positivity was very high attributable to the reasons discussed above, i.e., testing was restricted to high-risk symptomatic suspects. However, as the testing criteria became broad and more inclusive, the test positivity trend continued at low levels (<10%) albeit slightly higher at the height of the pandemic, coinciding with the increased levels of transmission.”

Query 17: “As the pandemic progressed, the testing of asymptomatic individuals could reflect the performance of the testing strategies. Certain states like Andhra Pradesh, Gujarat and Madhya Pradesh still continued to test a higher proportion of symptomatic individuals despite this fact .This shows that states followed different strategies for testing individuals despite regular revisions of ICMR testing strategies.” - Or that testing should be scaled up in this regions to avoid higher under-ascertainment of cases.

Response: It is likely that variable testing across states might have influenced the accurate ascertainment of total number of actual cases. But this ascertainment was not possible within the scope of the current analysis of the laboratory data. We have added a paragraph in the discussion section referring to this issue as follows.[Line 917-980]

“It is likely that the laboratory surveillance system suffered from a certain degree of under-ascertainment of cases. This could be attributed to various reasons. Firstly, testing was voluntary and done only among those who satisfied the eligibility criteria. Since >90% of the infections were asymptomatic, this testing strategy was likely to have missed them and secondly, the sensitivity of the testing kits ranged from 70 to 90%, which again would have led to misclassifying cases into false negatives. Evidence for this under-ascertainment comes from the infection-to-case ratios (ICR) estimated over time from the three rounds of nation-wide sero-surveys in India [20,41,42], which estimates the number likely infections undetected in the community for every case identified by the surveillance. However, the ICR decreased from 81.6 (in May, 2020) to 26 (in August, 2020) to 27 (in December, 2020) suggesting that the sensitivity of the surveillance improved over time.”

Query 18: “Although higher positivity indicates increased transmission, a detailed investigation on why certain states experience exceptionally high positivity needs further investigation” Refer that this may imply high levels of under-ascertainment, undetected cases.

Response: The sentence has been modified accordingly.[Line 797]

“Although higher positivity indicates increased transmission, a detailed investigation on why certain states experience exceptionally high positivity needs further investigation, including the possibility of under-ascertainment of cases.”

Query 19: Avoid repeating results in discussion. Do it only briefly to discuss relevant aspects.

Response: Deleted the following sentences which were repetition of results.

“, from merely 40,000 tests per day in the early phase of the pandemic to that of almost 1 million tests per day in 2021” 

“Missing or invalid data ranged from 0.01% to 0.7% across the study variables.”

Query 20: “One of the most important strategies in the fight against COVID-19 was more testing and more isolation” – could you clarify “more isolation”

Response: This is sentence was meant to refer to the test, trace, isolate and treat strategy that is recommended for the control of COVID-19 pandemic. To add more clarity the sentence has been revise to the following with a reference. [Line 802]

“One of the most important strategies in the fight against COVID-19 was increased testing and timely and appropriate isolation of positive individuals.[30–33]”

Query 21: “Although, all kinds of testing kits (like Rapid Antigen Tests and different brands of RT-PCR) with varying diagnostic validity were employed to detect the infection, we were unable to analyse the impact of this on the test positivity” Why was it not possible? Was “Test method” not registered in the database?

Response: The laboratories in the network used several brands of kits (both RT-PCR and RAT) and as such this information was available in the database. Therefore, we wish to submit that examining the role of brand of testing and how it affected the test results is beyond the scope of this paper.

Query 22: In the end of strengths and limitations it is of high importance to refer surveillance system sensitivity , potential high under-ascertainment, specially n regions with higher positivity and lower tests per capita. This should inform needs to upgrade surveillance.

Include references related to under-ascertainment/under-detection of cases/infection.

It is important to discuss, based on relevant references, serologic studies and others what was the level of under-ascertainment/under-detection of cases/infection during the whole period and in different periods.

Response: We agree with the reviewer. The following paragraph has been added in the limitations section.[Line 917-980]

“It is likely that the laboratory surveillance system suffered from a certain degree of under-ascertainment of cases. This could be attributed to various reasons. Firstly, testing was voluntary and done only among those who satisfied the eligibility criteria. Since >90% of the infections were asymptomatic, this testing strategy was likely to have missed them and secondly, the sensitivity of the testing kits ranged from 70 to 90%, which again would have led to misclassifying cases into false negatives. Evidence for this under-ascertainment comes from the infection-to-case ratios (ICR) estimated over time from the three rounds of nation-wide sero-surveys in India [20,41,42], which estimates the number likely infections undetected in the community for every case identified by the surveillance. However, the ICR decreased from 81.6 (in May, 2020) to 26 (in August, 2020) to 27 (in December, 2020) suggesting that the sensitivity of the surveillance improved over time.”

The following line has been added in the recommendation on upgrading the surveillance system to be more sensitive and reduce the level of under-ascertainment.[Line 1014]

“Ensuring a robust and sensitive surveillance system that can reduce the level of under-ascertainment of cases is absolutely essential to control transmission, better understand transmission dynamics, and changes in vaccine effectiveness and early detection on new variants of concern that may impact the future course of the pandemic.”

Query 23: Conclusions Include “positivity and test per capita and % test in symptomatic individual varied in different regions.

Response: We have added this to the conclusion section.[Line 984]

The sentence reads “2. Positivity and test per capita and % of tests in symptomatic individuals varied in different geographical regions,” 

Query 24: Recommendations: Consider including “ Regions with higher positivity, lower test per capita and higher % of symptomatic tested should improve test seeking behavior and access to testing as they have higher under-ascertainment/under-detection of cases/infection. Timeliness can be further improved with stronger and targeted communication campaigns and improving/facilitating access to testing and considering other broader screening strategies.

Response: We have added this to the recommendations. [Line 1010]

“Regions with higher positivity, lower test per capita and higher proportion of symptomatic tested should improve test seeking behaviour and access to testing. Timeliness of the surveillance system can be further improved with stronger and targeted communication campaigns, improving access to testing, and considering locally appropriate screening strategies.”

Query 25: Can be of relevance to consider a brief note in recommendations or discussion to the importance of guaranteeing high surveillance system sensitivity/ case ascertainment case detection to allow for control of transmission, understanding transmission dynamics and changes in vaccine effectiveness and early detection and research on new variants of concern that may impact the future. This is why surveillance improvement in COVID-19 is so relevant everywhere for the world.

Response: The following has been added to the recommendations section.[Line 1014]

“Ensuring a robust and sensitive surveillance system that can reduce the level of under-ascertainment of cases is absolutely essential to control transmission, better understand transmission dynamics, and changes in vaccine effectiveness and early detection on new variants of concern that may impact the future course of the pandemic.”

---

## [Decision Letter · Decision Letter 1]

29 Sep 2021

PONE-D-21-24568R1COVID-19 testing, timeliness and positivity from ICMR’s laboratory surveillance network in India: Profile of 176 million individuals tested and 188 million tests, March 2020 to January 2021PLOS ONE

Dear Dr. Murhekar,

Thank you for submitting your manuscript to PLOS ONE. After careful consideration, we feel that it has merit but does not fully meet PLOS ONE’s publication criteria as it currently stands. Therefore, we invite you to submit a revised version of the manuscript that addresses the points raised during the review process.

At the outset, I take this opportunity to thank you personally, and all the reviewers recognize their hard work. Two of the reviewers have expressed few more concerns, which I think need to be addressed by the authors. Please find the specific comments of the reviewers as attached.

We look forward to receiving your revised manuscript.

Kind regards,

Sandul Yasobant, PhD

Academic Editor

PLOS ONE

Journal Requirements:

Additional Editor Comments (if provided):

Copy editing by the professionals is recommended. 

Reviewers' comments:

Reviewer's Responses to Questions

**Comments to the Author**

1. If the authors have adequately addressed your comments raised in a previous round of review and you feel that this manuscript is now acceptable for publication, you may indicate that here to bypass the “Comments to the Author” section, enter your conflict of interest statement in the “Confidential to Editor” section, and submit your "Accept" recommendation.

Reviewer #1: (No Response)

Reviewer #3: All comments have been addressed

2. Is the manuscript technically sound, and do the data support the conclusions?

Reviewer #1: (No Response)

Reviewer #3: Yes

3. Has the statistical analysis been performed appropriately and rigorously? 

Reviewer #1: I Don't Know

Reviewer #3: Yes

4. Have the authors made all data underlying the findings in their manuscript fully available?

Reviewer #1: Yes

Reviewer #3: No

5. Is the manuscript presented in an intelligible fashion and written in standard English?

Reviewer #1: No

Reviewer #3: Yes

6. Review Comments to the Author

Reviewer #1: The language still needs editing. Especially the re-written/added paragraphs should be checked.

It should be clarified in the tables and figures whether the numbers refer to tested individuals or tests.

The age group 18-60 could be divided in smaller groups. 18-year old people are very different from 60-year old people. Even though the results would not change it would be reasonable to divide this group into at least two groups 18-40, 41-60.

Reviewer #3: Review 2

COVID-19 testing, timeliness and positivity from ICMR’s laboratory surveillance

network in India: Profile of 176 million individuals tested and 188 million tests, March

2020 to January 202

“India’s COVID-19 testing strategy evolved with the changing needs of the pandemic. In March 2020, when the testing resources were limited, only high-risk symptomatic individuals, international inbound travellers, high-risk contacts and patients with Severe Acute Respiratory Illness (SARI) were eligible for testing. During the later phases of the pandemic, the eligibility criteria was expanded to include, asymptomatic contacts, surveillance of symptomatic persons in containments zones, patients with Influenza Like Illness (ILI) and, migrant workers. More recently, patient-initiated testing was also added on where situations such as international travel, screening before undergoing medical/surgical procedures were also eligible for testing. A number of public-centric initiatives were taken by the government to provide free and wide access to testing. Persons who wanted testing were asked to contact the nearest public health care facility where testing was provided free of cost. Testing was carried out at homes of symptomatic patients/contacts of laboratory confirmed cases by healthcare workers during routine house-to-house fever surveillance. Mobile vans located at strategic places in major towns and cities also provided testing facilities to people who wanted to get tested. Apart from these government initiatives, many private labs and hospitals also provided fee-based testing.[2,3]”

It could be relevant to give some specific timings instead of “more recently”. If they varied too much between regions it should be refered.

Are there any other reasons (beyond broader testing) why 90% of tested people were not symptomatic? Could there be information bias (social desirability?) people may not report symptoms to the health professional who is collecting the data and testing?

“It is likely that the laboratory surveillance system suffered from a certain degree of under-ascertainment of cases. This could be attributed to various reasons. Firstly, testing was voluntary and done only among those who satisfied the eligibility criteria. Since >90% of the infections were asymptomatic, this testing strategy was likely to have missed them and secondly, the sensitivity of the testing kits ranged from 70 to 90%, which again would have led to misclassifying cases into false negatives. Evidence for this under-ascertainment comes from the infection-to-case ratios (ICR) estimated over time from the three rounds of nation-wide sero-surveys in India [20,41,42], which estimates the number likely infections undetected in the community for every case identified by the surveillance. However, the ICR decreased from 81.6 (in May, 2020) to 26 (in August, 2020) to 27 (in December, 2020) suggesting that the sensitivity of the surveillance improved over time.”

It is not clear that “Since >90% of the infections were asymptomatic, this testing strategy was likely to have missed them”. Is this correct? What doess it means. I though 90% of those tested were asymptomatic. If 90% of infections were asymptomatic this warrants further discussion as this is very different from what is found in many other countries.

“Evidence for this under-ascertainment comes from the infection-to-case ratios (ICR) estimated over time from the three rounds of nation-wide sero-surveys in India [20,41,42], which estimates the number likely infections undetected in the community for every case identified by the surveillance. However, the ICR decreased from 81.6 (in May, 2020) to 26 (in August, 2020) to 27 (in December, 2020) suggesting that the sensitivity of the surveillance improved over time.” – This is good information. Thank you.

“Persons who wanted testing were asked to contact the nearest public health care facility where testing was provided free of cost.” - Only if they meet the other criteria correct?

I understand it may be difficult but could you refer how many public health care facilities exist in the country or the average population and dimension of the area it serves? This is not absolutely necessary ofcourse but is relevant for context of accessibility to testing.

“Further, it was seen that the testing rate was higher in men, than in women across all states, in contrast to developed countries in the west.[22] Among those who were tested, the proportion of women was highest in Andhra Pradesh (46.4%) and lowest in Dadra and Nagar Haveli (23.4%). It has been previously shown that young women in certain states of India were less likely than men to practice key COVID-19 preventive behaviours.[23] Thus, such low frequency of testing behaviour among women was reflected in low prevalence among women as well.”

Consider changing “developed countries in the wes”to “high-income countries in the west”

“The overall percentage of symptomatic individual who were tested was lower in our study. This is due to two main reasons – firstly, the testing criteria was kept relatively broad and a large proportion of testing was carried out among asymptomatic individuals who were contacts of cases or picked up for random testing in containment areas [14,15] and secondly, data on symptoms may have been inaccurately reported, collected, or entered in the database.”

This is ok .

“An important indicator of the efficiency of a nationwide pandemic surveillance system is the timeliness of sample collection after an individual develops symptoms.[37,38] In our analysis, we found that in the initial phases of the pandemic the duration between symptom onset and sample collection was three days. In the subsequent phases, we found that the median duration was reduced to zero. This is an important finding because it means that the surveillance system became more responsive and individuals were becoming increasingly aware of the need to get tested immediately after they develop symptoms. An direct effect of this reduction in delay would be more effective isolation of symptomatic patients, early detection of possible contacts and their quarantine. This could have played an pivotal role in halting the progress of continued viral transmission [39], preventing the progress of infection to severe disease and reduction of complications and mortality.”

-The fact that In the subsequent phases, you found that the median duration from symptom onset to testing was reduced to zero needs to be discussed considering social desirability bias. It seems impossible that everyone is test in the exact day they begin symptoms. This may be a very relevant social phenomnenon for the future of the pandemic globally.

“However, other explanations include the possibility that high levels of under-ascertainment of cases could also be operating within the states.”

- This is likely. It should not seem as a remote possibility. Consider rephrasing.

“Although higher positivity indicates increased transmission, a detailed investigation on why certain states experience exceptionally high positivity needs further investigation, including the possibility of under-ascertainment of cases.”

Conside writing “the likely higher under-ascertainement of cases and its implications”.

“Ensuring a robust and sensitive surveillance system that can reduce the level of under-ascertainment of cases is absolutely essential to control transmission, better understand transmission dynamics, and changes in vaccine effectiveness and early detection on new variants of concern that may impact the future course of the pandemic”

This is very relevant. Thanks to the authors.

7. PLOS authors have the option to publish the peer review history of their article (what does this mean?). If published, this will include your full peer review and any attached files.

Reviewer #1: No

Reviewer #3: No

---

## [Author Response · Author response to Decision Letter 1]

18 Oct 2021

Responses to Reviewer's comments

Reviewer #1:

Query 1: The language still needs editing. Especially the re-written/added paragraphs should be checked.

Response: We thank the reviewer for this comment. We have now professionally edited the language to meet the journal standards. The changes made can be seen in the track-change document.

Query 2: It should be clarified in the tables and figures whether the numbers refer to tested individuals or tests.

Response: We thank the reviewer for this comment. We have changed the table and figure titles to clearly indicate individuals and tests as follows.

Table 1. COVID-19 testing, incidence and positivity by gender and age among tested individuals, India (March 2020 to January 2021)

Table 2. Characteristics of the COVID-19 tests carried out in the laboratory surveillance network, India (March 2020 to January 2021)

Fig 1. State-wise distribution of COVID-19 testing, incidence and positivity among tested individuals, India (March 2020 to January 2021)

Fig 2. Distribution of laboratory-confirmed COVID-19 cases by week and demographic characteristics, India (March 2020 to January 2021)

Fig 3. Distribution of laboratory-confirmed COVID-19 cases and percentage change by week and the state of residence, India (March 2020 to January 2021)

Fig 4. Weekly and cumulative incidence of laboratory-confirmed COVID-19 cases by the district of residence, India (March 2020 to January 2021)

Fig 5. Distribution of laboratory-confirmed COVID-19 cases by dates of symptom onset, sample collection and result confirmation, India (March 2020 to January 2021)

Fig 6. Trends of the characteristics of the COVID-19 tests, India (March 2020 to January 2021)

Table 3. Timeliness of the COVID-19 tests carried out in the laboratory surveillance network by scale-up phases and test results, India (March 2020 to January 2021)

Fig 7. Timeliness of the COVID-19 tests carried out in the laboratory surveillance network by scale-up phases and test results, India (March 2020 to January 2021)

Query 3: The age group 18-60 could be divided in smaller groups. 18-year old people are very different from 60-year old people. Even though the results would not change it would be reasonable to divide this group into at least two groups 18-40, 41-60.

Response: We thank the reviewer for this comment. We have added the indicators for the suggested age groups in all the relevant tables, figures and description.

Reviewer #3: 

Query 1: “India’s COVID-19 testing strategy evolved with the changing needs of the pandemic. In March 2020, when the testing resources were limited, only high-risk symptomatic individuals, international inbound travellers, high-risk contacts and patients with Severe Acute Respiratory Illness (SARI) were eligible for testing. During the later phases of the pandemic, the eligibility criteria was expanded to include, asymptomatic contacts, surveillance of symptomatic persons in containments zones, patients with Influenza Like Illness (ILI) and, migrant workers. More recently, patient-initiated testing was also added on where situations such as international travel, screening before undergoing medical/surgical procedures were also eligible for testing. A number of public-centric initiatives were taken by the government to provide free and wide access to testing. Persons who wanted testing were asked to contact the nearest public health care facility where testing was provided free of cost. Testing was carried out at homes of symptomatic patients/contacts of laboratory confirmed cases by healthcare workers during routine house-to-house fever surveillance. Mobile vans located at strategic places in major towns and cities also provided testing facilities to people who wanted to get tested. Apart from these government initiatives, many private labs and hospitals also provided fee-based testing.[2,3]”

It could be relevant to give some specific timings instead of “more recently”. If they varied too much between regions it should be refered.

Response: We thank the reviewer for this comment. The guideline for patient-initiated testing (also known as Test-on-demand) was issued in September, 2020 across the country. We do not have information on where the implementation of this guideline was uniform or not. The indicated line has been modified as given below.[Line 101]

“Since September 2020, patient-initiated testing was also added to the criteria, along with situations such as international travel, screening before undergoing medical/surgical procedures were also eligible for testing.” 

Query 2: Are there any other reasons (beyond broader testing) why 90% of tested people were not symptomatic? Could there be information bias (social desirability?) people may not report symptoms to the health professional who is collecting the data and testing?

Response: The percentage of tests that were carried out among symptomatic individuals was 7%. We attribute this mainly to the broad nature of the testing criteria used (including contact tracing, international travellers, migrant workers and health care workers). We are unable to comment on whether a social desirability bias was operating where patients were unwilling to disclose to symptoms. Having said that there is no evidence in literature from India to suggest that individuals would refrain from disclosing their symptoms at the time of testing. Also, the symptom information was collected at the time of testing and the individuals could have developed symptoms during their course of illness. 

Query 3: “It is likely that the laboratory surveillance system suffered from a certain degree of under-ascertainment of cases. This could be attributed to various reasons. Firstly, testing was voluntary and done only among those who satisfied the eligibility criteria. Since >90% of the infections were asymptomatic, this testing strategy was likely to have missed them and secondly, the sensitivity of the testing kits ranged from 70 to 90%, which again would have led to misclassifying cases into false negatives. Evidence for this under-ascertainment comes from the infection-to-case ratios (ICR) estimated over time from the three rounds of nation-wide sero-surveys in India [20,41,42], which estimates the number likely infections undetected in the community for every case identified by the surveillance. However, the ICR decreased from 81.6 (in May, 2020) to 26 (in August, 2020) to 27 (in December, 2020) suggesting that the sensitivity of the surveillance improved over time.”

It is not clear that “Since >90% of the infections were asymptomatic, this testing strategy was likely to have missed them”. Is this correct? What doess it means. I though 90% of those tested were asymptomatic. If 90% of infections were asymptomatic this warrants further discussion as this is very different from what is found in many other countries.

Response: We thank the reviewer for pointing this out. 

In general, the actual percentage of asymptomatic COVID-19 infection varies. According to a systematic review [https://www.pnas.org/content/118/34/e2109229118], 42.8% of COVID-19 affected individuals were asymptomatic at the time of testing and 35.1% were truly asymptomatic, in that they did not develop any symptom throughout the course of the infection. So, a significant proportion of the those COVID-19 infections were asymptomatic. In India also, studies [https://www.ijidonline.com/article/S1201-9712(21)00442-2/fulltext#secsect0085] have reported that asymptomatic COVID-19 infection was very prevalent and this is also affected by the emergence of new variants like Delta which have a lower proportion of asymptomatic infection. The general point that we are trying to make is that the surveillance system may have missed these asymptomatic infections. To reflect this more accurately in the manuscript, the sentence has been rephrased as follows.[Line 787]

“Since a significant proportion of the COVID-19 infections were likely asymptomatic, this testing strategy may have missed them, and secondly, the varying sensitivity of the testing kits would have led to misclassifying cases into false negatives.”

Query 4: “Evidence for this under-ascertainment comes from the infection-to-case ratios (ICR) estimated over time from the three rounds of nation-wide sero-surveys in India [20,41,42], which estimates the number likely infections undetected in the community for every case identified by the surveillance. However, the ICR decreased from 81.6 (in May, 2020) to 26 (in August, 2020) to 27 (in December, 2020) suggesting that the sensitivity of the surveillance improved over time.” – This is good information. Thank you.

Response: We thank the reviewer for this comment. 

Query 5: “Persons who wanted testing were asked to contact the nearest public health care facility where testing was provided free of cost.” - Only if they meet the other criteria correct?

Response: Yes, patients were tested as per the testing criteria.

Query 6: I understand it may be difficult but could you refer how many public health care facilities exist in the country or the average population and dimension of the area it serves? This is not absolutely necessary of course but is relevant for context of accessibility to testing.

Response: We thank the reviewer for this comment. It would not be possible within the scope of this paper to provide an assessment of the number of health facilities in a country as large as India, especially given the diverse nature of the prevailing health care delivery system. However, we have given details of the number of COVID-19 testing laboratories in each state in Table S2. By no means this can be taken a proxy for access to testing because these laboratories were linked to several health facilities who provided the samples collected from the patients.

Query 7: “Further, it was seen that the testing rate was higher in men, than in women across all states, in contrast to developed countries in the west.[22] Among those who were tested, the proportion of women was highest in Andhra Pradesh (46.4%) and lowest in Dadra and Nagar Haveli (23.4%). It has been previously shown that young women in certain states of India were less likely than men to practice key COVID-19 preventive behaviours.[23] Thus, such low frequency of testing behaviour among women was reflected in low prevalence among women as well.” Consider changing “developed countries in the wes”to “high-income countries in the west”

Response: The sentences has been modified as follows.[Line 539]

“Further, it was seen that the testing rate was higher in men, than in women across all states, in contrast to high-income countries in the west.[22]”

Query 8: “The overall percentage of symptomatic individual who were tested was lower in our study. This is due to two main reasons – firstly, the testing criteria was kept relatively broad and a large proportion of testing was carried out among asymptomatic individuals who were contacts of cases or picked up for random testing in containment areas [14,15] and secondly, data on symptoms may have been inaccurately reported, collected, or entered in the database.” This is ok.

Response: We thank the reviewer for this comment. 

Query 9: “An important indicator of the efficiency of a nationwide pandemic surveillance system is the timeliness of sample collection after an individual develops symptoms.[37,38] In our analysis, we found that in the initial phases of the pandemic the duration between symptom onset and sample collection was three days. In the subsequent phases, we found that the median duration was reduced to zero. This is an important finding because it means that the surveillance system became more responsive and individuals were becoming increasingly aware of the need to get tested immediately after they develop symptoms. An direct effect of this reduction in delay would be more effective isolation of symptomatic patients, early detection of possible contacts and their quarantine. This could have played an pivotal role in halting the progress of continued viral transmission [39], preventing the progress of infection to severe disease and reduction of complications and mortality.”

-The fact that In the subsequent phases, you found that the median duration from symptom onset to testing was reduced to zero needs to be discussed considering social desirability bias. It seems impossible that everyone is test in the exact day they begin symptoms. This may be a very relevant social phenomnenon for the future of the pandemic globally.

Response: Not everyone got tested on the same day of symptom onset. This value only represents the median value, which means that at least 50% of the individuals got tested on the day of symptom onset. A large proportion of symptomatic patients still did not get tested on the same day. There is no evidence in Indian literature to suggest that individuals would not disclose their symptoms while testing due to social desirability bias. Therefore, we would be unable to comment on this hypothesis. We have added the following sentence to reflect this.[Line 677]

“In subsequent phases, at least 50% of individuals got tested at symptoms onset, with another 25% delaying their testing by 3 to 4 days. This interval is important because it indicates that the surveillance system became more responsive with time, and individuals became increasingly aware that they needed testing immediately after developing symptoms.”

Query 10: “However, other explanations include the possibility that high levels of under-ascertainment of cases could also be operating within the states.”

- This is likely. It should not seem as a remote possibility. Consider rephrasing.

Response: The sentence has been rephrased as below.[Line 565]

“However, the possibility of high levels of under-ascertainment of cases within the states cannot be overstated.”

Query 11: “Although higher positivity indicates increased transmission, a detailed investigation on why certain states experience exceptionally high positivity needs further investigation, including the possibility of under-ascertainment of cases.”

Conside writing “the likely higher under-ascertainement of cases and its implications”.

Response: The rephrased sentence now reads:[Line 622]

“Although higher positivity indicates increased transmission, a detailed investigation on why certain states experience exceptionally high positivity is needed focussing on the high under-ascertainment of cases and its implications.”

Query 12: “Ensuring a robust and sensitive surveillance system that can reduce the level of under-ascertainment of cases is absolutely essential to control transmission, better understand transmission dynamics, and changes in vaccine effectiveness and early detection on new variants of concern that may impact the future course of the pandemic”

This is very relevant. Thanks to the authors.

Response: We thank the reviewer for this comment.

---

## [Decision Letter · Decision Letter 2]

22 Nov 2021

COVID-19 testing, timeliness and positivity from ICMR’s laboratory surveillance network in India: Profile of 176 million individuals tested and 188 million tests, March 2020 to January 2021

PONE-D-21-24568R2

Dear Dr. Murhekar,

We’re pleased to inform you that your manuscript has been judged scientifically suitable for publication and will be formally accepted for publication once it meets all outstanding technical requirements.

Kind regards,

Sandul Yasobant, PhD

Academic Editor

PLOS ONE

Additional Editor Comments (optional):

Dear Author,

Congratulation for this piece of work. At the outstate of this, I must thank you to each authors for addressing all the comments and required revisions done as per suggestions of reviewers. May I request you to incorporate the minor editorial change/restructuring of sentence as suggested by Reviewer-3. Kindly consider the same during type setting.

Thanks and all the very best.

Reviewers' comments:

Reviewer's Responses to Questions

**Comments to the Author**

1. If the authors have adequately addressed your comments raised in a previous round of review and you feel that this manuscript is now acceptable for publication, you may indicate that here to bypass the “Comments to the Author” section, enter your conflict of interest statement in the “Confidential to Editor” section, and submit your "Accept" recommendation.

Reviewer #1: All comments have been addressed

Reviewer #3: All comments have been addressed

2. Is the manuscript technically sound, and do the data support the conclusions?

Reviewer #1: Yes

Reviewer #3: Yes

3. Has the statistical analysis been performed appropriately and rigorously? 

Reviewer #1: I Don't Know

Reviewer #3: Yes

4. Have the authors made all data underlying the findings in their manuscript fully available?

Reviewer #1: Yes

Reviewer #3: No

5. Is the manuscript presented in an intelligible fashion and written in standard English?

Reviewer #1: Yes

Reviewer #3: Yes

6. Review Comments to the Author

Reviewer #1: (No Response)

Reviewer #3: Thank you for adressing the issues.

I believe the manuscript should now be accepted.

A very minor suggestion for change

“Ensuring a robust and sensitive surveillance system that can reduce the

level of under-ascertainment of cases is absolutely essential to control transmission,

better understand transmission dynamics, and changes in vaccine effectiveness and

early detection on new variants of concern that may impact the future course of the

pandemic”

to

“Ensuring a robust and sensitive surveillance system that can reduce the

level of under-ascertainment of cases is absolutely essential to control transmission,

better understand transmission dynamics, changes in vaccine effectiveness !!and in immunity from previous infection!! and

early detection on new variants of concern that may impact the future course of the

pandemic”

I have nothing further to add.

7. PLOS authors have the option to publish the peer review history of their article (what does this mean?). If published, this will include your full peer review and any attached files.

Reviewer #1: No

Reviewer #3: No

---

## [Editor Report · Acceptance letter]

24 Nov 2021

PONE-D-21-24568R2 

COVID-19 testing, timeliness and positivity from ICMR’s laboratory surveillance network in India: Profile of 176 million individuals tested and 188 million tests, March 2020 to January 2021 

Dear Dr. Murhekar:

I'm pleased to inform you that your manuscript has been deemed suitable for publication in PLOS ONE. Congratulations! Your manuscript is now with our production department. 

Kind regards, 

on behalf of

Dr. Sandul Yasobant 

Academic Editor

PLOS ONE